# mTOR dysregulation induces IL-6 and paracrine AT2 cell senescence impeding lung repair in lymphangioleiomyomatosis

Roya Babaei-Jadidi [1] ✉, Debbie Clements[1], Yixin Wu [2], Ken Chen [2], Suzanne Miller[1], Manuela Platé [3], Kyungtae Lim [4,8], Ryan Rue [5], Vera P. Krymskaya [5], Rachel Chambers [3], Emma L. Rawlins [4,6], Yan Xu [7] & Simon R. Johnson [1] ✉

Lymphangioleiomyomatosis (LAM) is a rare disease of women in which *TSC2* deficient 'LAM cells' with dysregulated mTOR signalling and recruited fibroblasts form nodules causing lung cysts and respiratory failure. We examine if mTOR dysregulation can induce senescence and impair the response to lung injury in LAM. The senescence markers p21, p16 and the SenMayo gene set are increased in LAM lungs and colocalise with alveolar type 2 cells. LAM models induce mTOR dependent senescence in alveolar type 2 cell organoids in vitro and in vivo. IL-6 produced by LAM cells, induces p16 and p21 in alveolar type 2 cells, inhibits epithelial wound resolution and is related to lung function in LAM patients. Rapamycin and the IL-6 receptor antagonist Tocilizumab reduce alveolar type 2 cell organoid p21 accumulation and Tocilizumab enhances epithelial wound repair. Targeting IL-6 signalling in parallel with mTOR inhibition, may reduce lung damage in LAM.

Lymphangioleiomyomatosis (LAM) is a rare multisystem disease that almost exclusively affects women. Most symptoms are caused by the progressive accumulation of lung cysts, which lead to pneumothorax, dyspnoea and respiratory failure. The disease may also be associated with lymphatic obstruction and a propensity to develop the benign mesenchymal tumour, angiomyolipoma[1]. LAM can occur both sporadically and also in women with the genetic disease tuberous sclerosis complex (TSC) where LAM is a major cause of disability and death in adults[2]. The key pathological feature is the LAM cell, a clone of cells with loss of *TSC* gene function, most commonly *TSC2*. *TSC* gene loss results in dysregulated mechanistic target of rapamycin (mTOR) signalling, leading to LAM cell expansion, migration, protection from cell death and altered metabolism[1]. LAM cells in the lung parenchyma

proliferate locally and attract wild-type cells forming 'LAM nodules'; clusters of cells comprising LAM associated fibroblasts (LAF), lymphatic endothelial cells, T cells and mast cells analogous to a tumour stroma[3–6]. As the disease progresses, LAM nodules become more complex, increasingly recruiting wild-type cells, which is associated with falling lung function[7].

One of the striking features of LAM is the unusual cystic destruction of the lung parenchyma, with lung cysts present even when LAM nodules are sparse[7]. The mechanism of lung damage is unknown although LAM nodules express matrix degrading proteases including cathepsin K[8] and matrix metalloproteinases[9] which are thought to cause lung cyst formation. In addition, knockdown of *TSC2* in mesenchyme also results in alveolar enlargement in mice, which is

[1]Centre for Respiratory Research, School of Medicine, University of Nottingham, Nottingham, UK. [2]Neonatology and Pulmonary Biology, Perinatal Institute, Cincinnati Children's Hospital Medical Center, Cincinnati, USA. [3]Centre for Inflammation and Tissue Repair, University College London, London, UK. [4]The Wellcome Trust/Cancer Research UK Gurdon Institute, University of Cambridge, Cambridge, UK. [5]Division of Pulmonary and Critical Care Medicine, Perelman School of Medicine, University of Pennsylvania, Pennsylvania, USA. [6]Department of Physiology, Development and Neuroscience, University of Cambridge, Cambridge, UK. [7]Divisions of Pulmonary Biology and Biomedical Informatics, Perinatal Institute, Cincinnati Children's Hospital Medical Center, University of Cincinnati College of Medicine, Cincinnati, USA. [8]Present address: Department of Life Sciences, Korea University, 145 Anam-Ro, Seoungbuk-Gu, Seoul, South Korea. ✉e-mail: roya.babaei-jadidi@nottingham.ac.uk; simon.johnson@nottingham.ac.uk

dependent on Wnt pathway activation[10]. Lung repair after injury is characterised by the expansion of alveolar type 2 (AT2) cells, the resident alveolar stem cell, which differentiate to replace damaged alveolar type 1 (AT1) cells to repair the gas exchange surface of alveoli. Single-cell RNA sequencing (scRNAseq) of human lung injury and organoid models of the repair process have revealed cells with both AT1 and AT2 markers. These intermediate cells, termed 'pre-alveolar type-1 transitional cell state' (PATS)[11], have also been identified in single cell analyses and dual label immunohistochemistry in LAM[12]. PATS are unable to differentiate into AT1 cells and may contribute to impaired repair in lung disease[13] and their presence in LAM suggests there may be a co-existent failure of lung repair.

Senescence is a state of irreversible cell cycle arrest, which, in health, prevents the replication of damaged cells, suppressing tumorigenesis and limiting scar formation. Senescence occurs due to repeated cell division resulting in telomere shortening (replicative senescence) or by DNA damage resulting from reactive oxygen species (ROS) or other stressors including cigarette smoke[14]. DNA damage activates p53 and p21, and other stressors can directly activate p16, both causing cell cycle arrest[15]. Importantly, senescent cells remain metabolically active and secrete pro-inflammatory cytokines, chemokines and proteases. This senescence-associated secretory phenotype (SASP) acts to induce senescence in neighbouring cells[16,17]. The accumulation of senescent cells in disease and aging, and the pro-inflammatory SASP result in reduced organ repair, altered immune function and accelerated tissue damage in response to injurious stimuli. Activation of mTORC1, either by ROS, adenosine monophosphate kinase (AMPK) inhibition or genetic loss of PTEN or *TSC*1/2, is central to senescence; downregulating sirtuins 1 and 6 to both suppress DNA repair and induce the SASP via NFκB signalling[18].

We therefore hypothesised that mTOR dysregulation in LAM might induce senescence of LAM nodule components and via the SASP induce senescence in surrounding AT2 cells impairing lung repair processes. We found that LAM cells secreted IL-6 in an mTOR dependent manner which was associated with lung function loss and disease activity. LAM cells and IL-6 induced senescence in LAFs within LAM nodules and in adjacent AT2 cells, with senescent cells progressively accumulating with advancing disease. IL-6 signalling was associated with delayed epithelial repair in vitro which could be restored using the IL-6 receptor antagonist Tocilizumab, suggesting that adding IL-6 suppression to mTOR inhibition may improve outcomes in LAM.

## Results

### Markers of senescence are present in LAM lungs

We first examined gene expression of the canonical senescence markers the cyclin dependent kinase (CDK) inhibitors *CDKN1A* (p21) and *CDKN2A* (p16), using the LAM cell atlas, an open-source data interface comprising single cell RNA sequencing data from 13 lung transplant donors[19]. In these advanced disease samples, p21 was ubiquitously expressed including in *TSC* null LAM^core cells. Within the alveolar epithelial cell population, p21 was most enriched in AT1/AT2 transitional cells (PATS). p16 expression was lower than p21 across all cell types, although >10% of LAM^CORE cells were p16 positive. Senescence markers were also significantly increased in macrophage subsets, mesothelial and ciliated secretory cells (Fig. 1A).

As p21 and p16 proteins are mostly regulated by ubiquitin mediated degradation rather than transcription, we used immunohistochemistry to examine p16 and p21 proteins in 21 LAM lung and three control lung samples. p21 and p16 were expressed in LAM lung nodules (Fig. 1B and supplementary and Figs. 1, 2). We next examined the activity of SAβgal in a murine homograft LAM model. Immuno-edited *Tsc2* null murine renal tumour cells were injected into the tail veins of immunocompetent C57BL/6 mice with control animals undergoing sham injections[20]. Tissues were harvested after four and eight weeks. Tumour nodules were visible in *Tsc2* null homografts and SAβgal

activity in whole lung lysates was greater than two-fold higher than control animals after eight weeks (Fig. 1C).

### LAM nodules contain senescent cells

To characterise the senescent cell types in the LAM nodule stroma we performed dual immunostaining for p21 and p16 with the LAM cell markers the target of anti-melanoma antibody PNL2 (henceforth PNL2) and GP100 respectively and MFAP5 and TCF21 to identify LAFs (Fig. 2A and supplementary Figs. 3, 4). p21 was present in LAM nodules compared with control lungs, p16 tended to be more common in LAM but not significantly so. Only occasional cells co-expressed p21 or p16 proteins and the LAM cell markers (Fig. 2A, B, and supplementary Fig. 3). MFAP5 is not expressed by LAM or inflammatory cells[21] but was colocalised with p16 in LAM nodules in early disease (Supplementary Fig. 4A), however, its extensive extracellular presence made it unsuitable for quantification. TCF21, a marker expressed by fibroblasts and LAM^core cells[22] showed significant co-localisation with p16 over and above that of PNL2 (Fig. 2A, B and supplementary Fig. 4B). Dual staining of p16 and the macrophage marker CD68 revealed a small population of CD68/p16 expressing cells with the majority of p16 expressed by CD68 negative cells although not within LAM nodules (Supplementary Fig. 5). In the LAM-related renal tumour angiomyolipoma, thought to differentiate from the same *TSC*^/- cell clone as LAM cells, 3.8% (SD. 0.84) of all cells expressed p21 and 4.2% (2.0) expressed p16[23] (Supplementary Fig. 6).

To understand the processes within the LAM nodule we used laser capture micro-dissection to isolate LAM nodules from lung tissue from 19 women with LAM and analysed gene expression by RNA sequencing (Supplementary table 1). 481 individual genes were upregulated in LAM nodules and 500 downregulated compared with healthy lung tissue (Fig. 2C and supplementary table 2). Pathway analysis showed alterations in processes known to underlie LAM biology including mTOR signalling, autophagy and hypoxia. In addition, there was a strong extracellular matrix signal including ECM organisation, ECM receptor interaction, TGFβ response and integrin signalling. Genes associated with apoptosis and IL-18 signalling were also increased (Fig. 2D).

### AT2 cells are senescent in LAM

We hypothesised that mTOR dysregulated cells within LAM nodules may induce senescence in neighbouring cells including the AT2 population. In LAM, AT2 cells are observed surrounding LAM nodules and lung cysts in addition to their normal alveolar niche[12,24]. Dual immunohistochemical staining in 15 LAM and five control lung tissues for the AT2 cell marker pro-surfactant protein C (pro-SPC) and either p21 or p16, showed healthy control AT2 cells expressed neither p16 or p21, whereas both LAM nodule-associated and parenchymal AT2 cells expressed p16 and p21 proteins (Fig. 3A and supplementary Fig. 7).

As senescence is a complex response, which cannot be robustly represented by a single marker we examined a validated panel of 125 genes responsive to senescence in a range settings[25]. The SenMayo gene signature was significantly increased in LAM derived AT2 cells from the LAM cell atlas when compared with normal AT2 cells (Fig. 3B and supplementary table 3). Examining pathways associated with the differentially expressed genes in LAM AT2 cells, we observed evidence of cytokine stimulation consistent with SASP activity, senescence, apoptosis and Wnt signalling (Fig. 3C).

To determine how senescence develops over time we quantified senescent cell types in patients at different disease stages stratified by their lung function deficit. The most common senescent cell type was p16/TCF21 double positive cells (senescent mesenchymal cells) which increased with worsening disease. Senescent AT2 cells accumulated from early disease, approaching 10% of AT2 cells in moderate and advanced disease. PNL2/p16 double positive LAM cells represented only 5% of LAM cells only in the most severe disease (Fig. 3D).

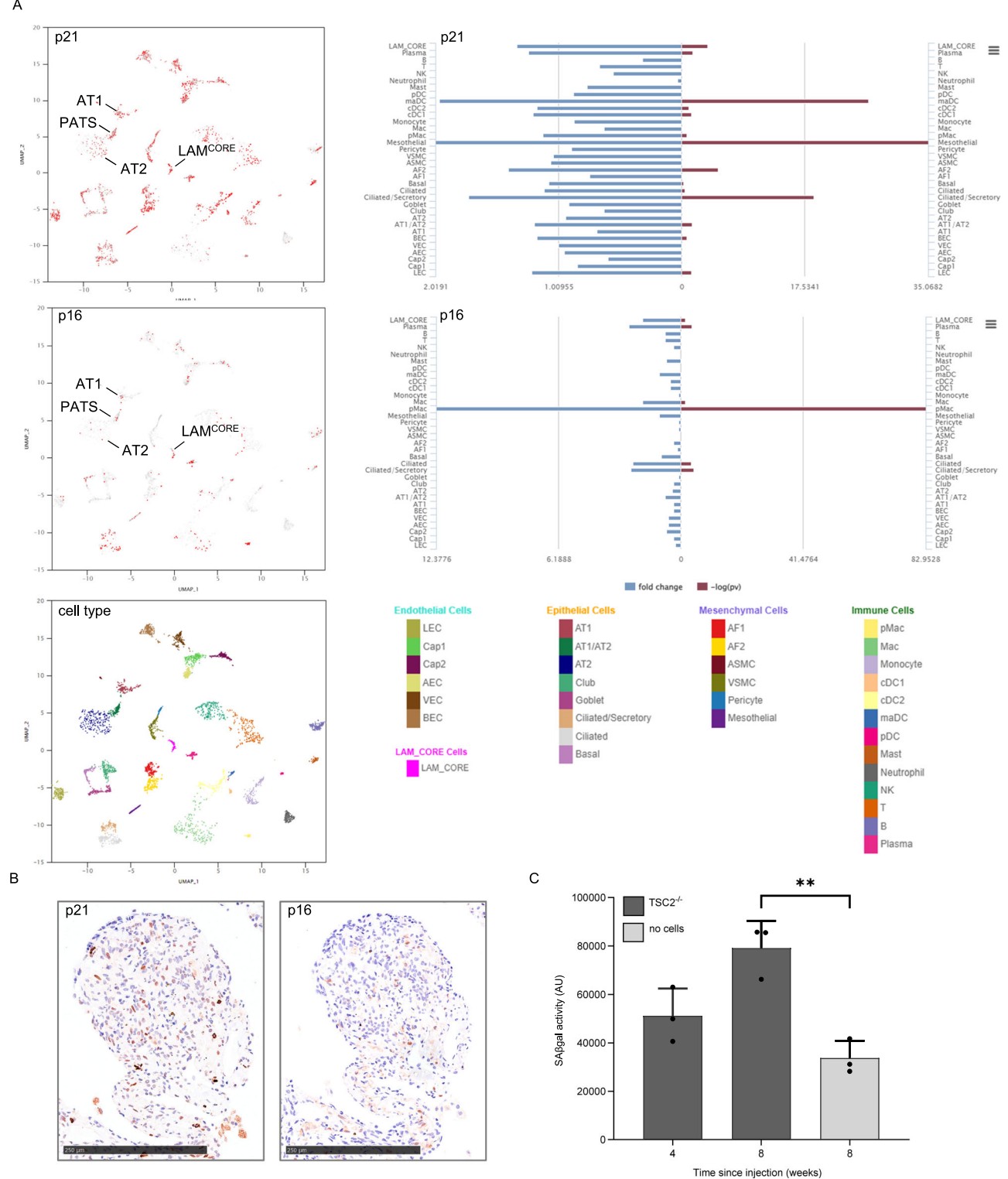

**Fig. 1 | Markers of senescence in LAM lung. A** Single cell RNA sequencing of LAM lung tissues showing expression of p16 and p21 in LAM lung populations using the LAM cell atlas[19]. Left panels show expression of positive cells for p21 and p16 in each cluster, right panels show the fold change and significance for LAM. Lower panel shows cell types present in LAM lung samples. AT1 = alveolar type 1 cells, AT2 = alveolar type 2 cells, PATS = pre-alveolar type-1 transitional cell state. **B** Immunostaining of LAM nodules for p16 and p21 protein, representative images of 21 LAM and 3 control lung sections. **C** Senescence associated beta galactosidase (SAβgal) activity in albino C57BL/6 mouse lungs after *TSC2*-/- cell injection or control (no cells). Graph shows mean (SD) of three animals per group and three technical replicates. 8-week points compared by unpaired two-tailed Student's t-test **p = 0.004. Source data are provided as a Source Data file.

To further understand the temporal evolution of these changes, their association with mTORC1 dysregulation and response to mTORC1 inhibition, we used the *Tsc2* null murine homograft model described above. *Tsc2* null murine renal tumour TTJ cells or saline (sham) were injected into the tail vein of albino C57BL/6 mice, two weeks later, animals were treated with the mTORC1 inhibitor rapamycin or vehicle. SAβgal/SPC, p21/SPC and p16/SPC dual positive AT2 cells were present at four weeks and increased after eight weeks

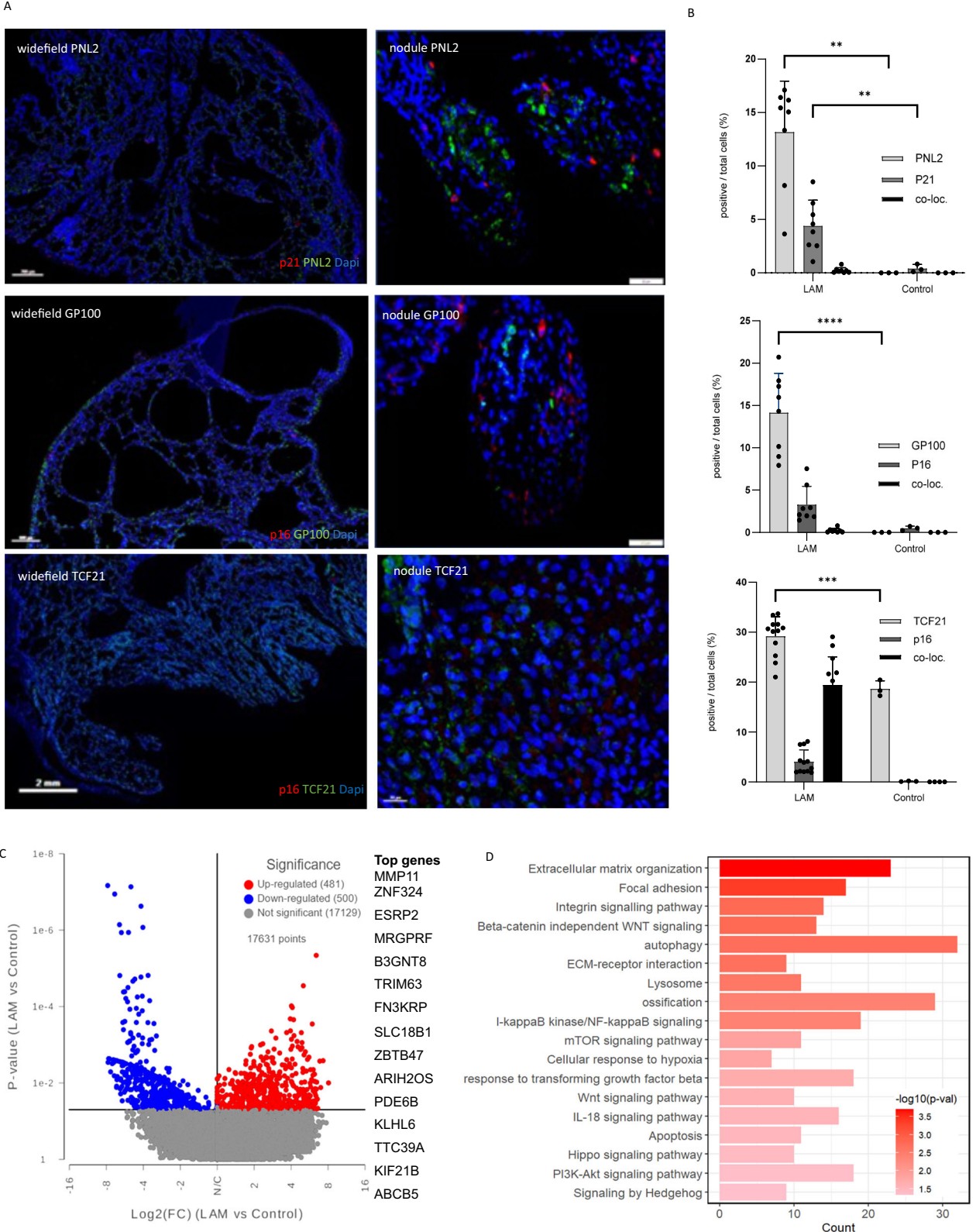

(Fig. 4A, B). BAL and serum IL-6 levels also increased over time (Fig. 4C), and although rapamycin treatment normalised IL-6 levels, it did not substantially affect p16, p21 or SAβgal expression (Fig. 4A, B). In a second animal model, *Tsc2* was deleted in mesenchymal cells using a *Tbx4*^Cre*Tsc*2KO mouse, which develops airspace enlargement and lung cysts[10]. Animals were treated with rapamycin or vehicle from birth to 20 weeks of age. *Tbx4*^Cre*Tsc*2KO mice developed airspace enlargement

associated with SAβgal/SPC, p21/SPC and p16/SPC dual positive AT2 cells. Rapamycin reduced airspace enlargement and almost completely abrogated SAβgal, p16 and p21 expression (Fig. 5A, B and supplementary Fig. 8, 9).

Although mTORC1 inhibitor treated LAM tissue is rarely accessible, we were able to examine mTORC1 suppression and AT2 cell senescence in two human LAM samples. We compared a panel of

**Fig. 2 | Senescence in LAM nodules. A** Representative widefield and close up images of dual label immunohistochemical staining of p16 and p21 and the LAM cell markers PNL2 and GP100 respectively and the mesenchymal cell marker TCF21. **B** Quantification of senescence and LAM cell markers in LAM (*n* = 8, ≥3 replicates) and control lungs (*n* = 3, ≥3 replicates). Graphs show the mean (SD) percentage of positive, compared with all cells in the region of interest and the percentage of cells expressing both markers (co-loc) analysed by 2-way ANOVA. Source data and exact *p*-values are provided as a Source Data file. **p < 0.01, ****p < 0.0001. **C** Volcano plot showing differentially expressed genes assessed by RNA sequencing in LAM nodules isolated by laser capture micro-dissection from 19 women with LAM compared with three healthy control lungs. The 15 most significantly increased genes are listed. Differentially expressed genes (DEG) between LAM vs control were identified using DESeq2[41], which performs a two-sided test. The y-axis shows *p* value (<0.05) and the x-axis shows Log2(FC) (FC > 1.5). FDR was calculated but not used for the DEG selection due to the high sample variations. **D** Pathway analysis of laser captured LAM nodules showing the most strongly increased pathways. **p < 0.01, ****p < 0.0001. Toppgene suite was used for gene sets and pathway enrichment analysis, *p* value is calculated using a Fisher's inverse chi-square method[48], a False Discovery Rate of <10% was applied.

senescence associated genes in control lung AT2 cells, and AT2 cells from rapamycin naive women with LAM and a single LAM patient treated with rapamycin using single cell sequencing. AT2 cell senescence associated genes were increased in LAM and were mostly normalised in the patient treated with rapamycin (Fig. 5C). However, examining a single LAM lung transplant sample obtained from a patient with advanced LAM taking the mTOR inhibitor Everolimus, we observed persisting p16 protein expressing AT2 and stromal cells (Fig. 5D). Although based on only two patients, the data are consistent with senescence being mTOR dependent in LAM and irreversible once established.

### SASP mediated senescence induction is mTOR dependent
To study the interactions between LAM nodule components and the alveolar epithelium we established a series of in vitro co-cultures incorporating LAM derived *TSC2* null 621-101 or control *TSC2* add-back 621-103 cells with primary LAFs isolated from LAM lung tissue and characterised senescence in low serum co-cultures over 14 days. LAF / 621 cell interactions induced a senescent phenotype characterised by enlarged flattened cells with loss of the proliferation marker nuclear Ki67, increased numbers of lysosomes, SAβgal activity, p16, p21 protein expression and nuclear expression of p16 (Fig. 6A–C, and supplementary Figs. 10, 11A). Using SAβgal, p21 and p16 proteins as markers of the senescent phenotype we showed that both LAM cells and LAFs could mutually induce senescence in the other cell type. IL-6 alone could enhance p16 and p21 expression in LAM cells (Fig. 6B, C). Senescence was mTOR dependent, with *TSC2* null 621-101 cells having higher levels of SAβgal than *TSC2* addback 621-103 cells and senescence induction by 621-101 cells resulting in greater SAβgal expression in LAFs than 621-103 cells (Fig. 6D). Bulk RNA sequencing of the LAM cells from co-cultures showed an induction of senescence-initiating genes including *p16, p21, TP53, RBL1, RBL2* in a *TSC2* dependent manner (Supplementary Fig. 11B).

We examined the effect of the mTORC1 inhibitor rapamycin and the senolytic drugs Dasatinib (Src kinase inhibitor) and Navitoclax (Bcl-2, Bcl-XL and Bcl-w inhibitor) on SAβgal activity in combinations of *TSC2* null and addback cells, LAFs and lung epithelial derived A549 cells. In LAFs, the *TSC2* null dependent induction of SAβgal activity was inhibited by rapamycin but not Dasatinib or Navitoclax (Fig. 6E). In epithelial cells, rapamycin blocked SAβgal activity induced by the presence of *TSC2* null LAM cell co-cultures but the addition of LAFs to the A549/LAM 621-101 co-cultures induced SAβgal activity that was resistant to rapamycin. Dasatinib reduced epithelial SAβgal activity in the *TSC2* null 621-101/A549 co-cultures but had no effect upon the *TSC2* addback 621-103/LAF/A549 co-cultures (Fig. 6F).

### LAM nodule cell interactions drive AT2 cell behaviour
To model these interactions using primary AT2 cells and the role of LAM cell mTOR dysregulation on senescence induction, we made 3D LAM spheroids (either *TSC2* null 621-101 or *TSC2* add-back 621-103 cells co-cultured with LAFs) which were incubated with embryo derived AT2 cell organoids (Fig. 7A). Control AT2 organoids increased in size steadily over 14 days, AT2 cell organoids exposed to individual LAF or *TSC2* addback 621-103 cells reduced organoid growth whereas *TSC2*

null 621-101 cells induced rapid initial AT2 organoid growth which plateaued early (Fig. 7B). Both *TSC2* null and addback 621 cell / LAF spheroids tended to inhibit AT2 organoid growth over two weeks, although not significantly so (Fig. 7C). To determine if this potential growth reduction was associated with the induction of senescence, organoids were immunostained for p16 and p21 proteins. Control organoids grown without LAM spheroids had low levels of expression of both proteins. Both p16 and p21 were induced in the presence of *TSC2* null 621-101 containing spheroids, whereas *TSC2* addback 621-103 containing spheroids had a smaller and non-significant effect on p16 and p21 induction (Figs. 7D, E and supplementary Fig. 12).

### Inhibition of IL-6 signalling facilitates epithelial wound repair processes and AT2 cell senescence
To understand the signalling events underlying AT2 cell senescence, we used upstream regulator analysis of scRNAseq data which predicted activation of AT2 cell oestrogen receptor β (ESR2), the interleukin 6 receptor (IL6R), toll like receptor 7 (TLR7), TNF receptor superfamily 1 A (TNFSF1A) and insulin dependent growth factor receptor 1 (IGF1R) (Fig. 8A). Of importance to senescence induction, IL6R, TNFSF1A, IGF1R and ESR2 are upstream of *CDKN1A* / p21 (Table 1).

IL-6 is a canonical SASP protein and activation of the IL-6 receptor suggests its involvement in the senescence response in LAM. To investigate this further, we first examined IL-6 protein secretion by LAM related cells.

*TSC2*$^{-/-}$ 621-101 cells produced IL-6 whereas *TSC2*$^{+/+}$ 621-103 cells expressed only minimal IL-6. LAFs produced IL-6 whereas lung epithelial derived A549 cells did not. Co-cultures of 621-101 cells and LAFs, matched for total cell number, produced more IL-6 than either cell type alone (Supplementary Fig. 13). To determine if IL-6 signalling could impact on lung repair we treated scratch wounded alveolar epithelial A549 cells with recombinant IL-6. IL-6 (10 ng/ml) significantly inhibited scratch wound resolution over 24 hours (Fig. 8B). Repair of scratch wounded A549 cells incubated with serum-containing conditioned media from *TSC2*$^{-/-}$ 621-101 cell / LAF spheroids was markedly enhanced by Tocilizumab which had a lesser but still significant effect on wound healing of *TSC2*$^{+/+}$ 621-103 / LAF spheroid conditioned medium (Fig. 8C).

We next analysed if IL-6 was part of the senescence response in AT2 cells, and how this related to mTOR signalling. IL-6 induced p16, and to a lesser extent p21 over 14 days in AT2 cell organoids. 621-101 cells induced p16 and SAβgal whereas the 621-103 *TSC2* addback cells had a smaller effect on p16 and did not induce SAβgal. 621-101/LAF co-cultures strongly induced SAβgal, p21 and p16. 621-103/LAF co-cultures had lesser effects on p21 and p16 induction and had no effect on SAβgal (Supplementary Fig. 12). In AT2 cell organoids exposed to *TSC2*$^{-/-}$ 621-101/LAF spheroids over 14 days, both rapamycin (10 ng/ml) and Tocilizumab (30 ng/ml) reduced the accumulation of p21 induced by LAM spheroids, the combination of rapamycin and Tocilizumab, further reduced p21 protein. A similar trend was seen for p16 (Fig. 8D, E).

To determine if IL-6 is increased in patients with LAM, we compared serum IL-6 in healthy control women, 88 women with LAM

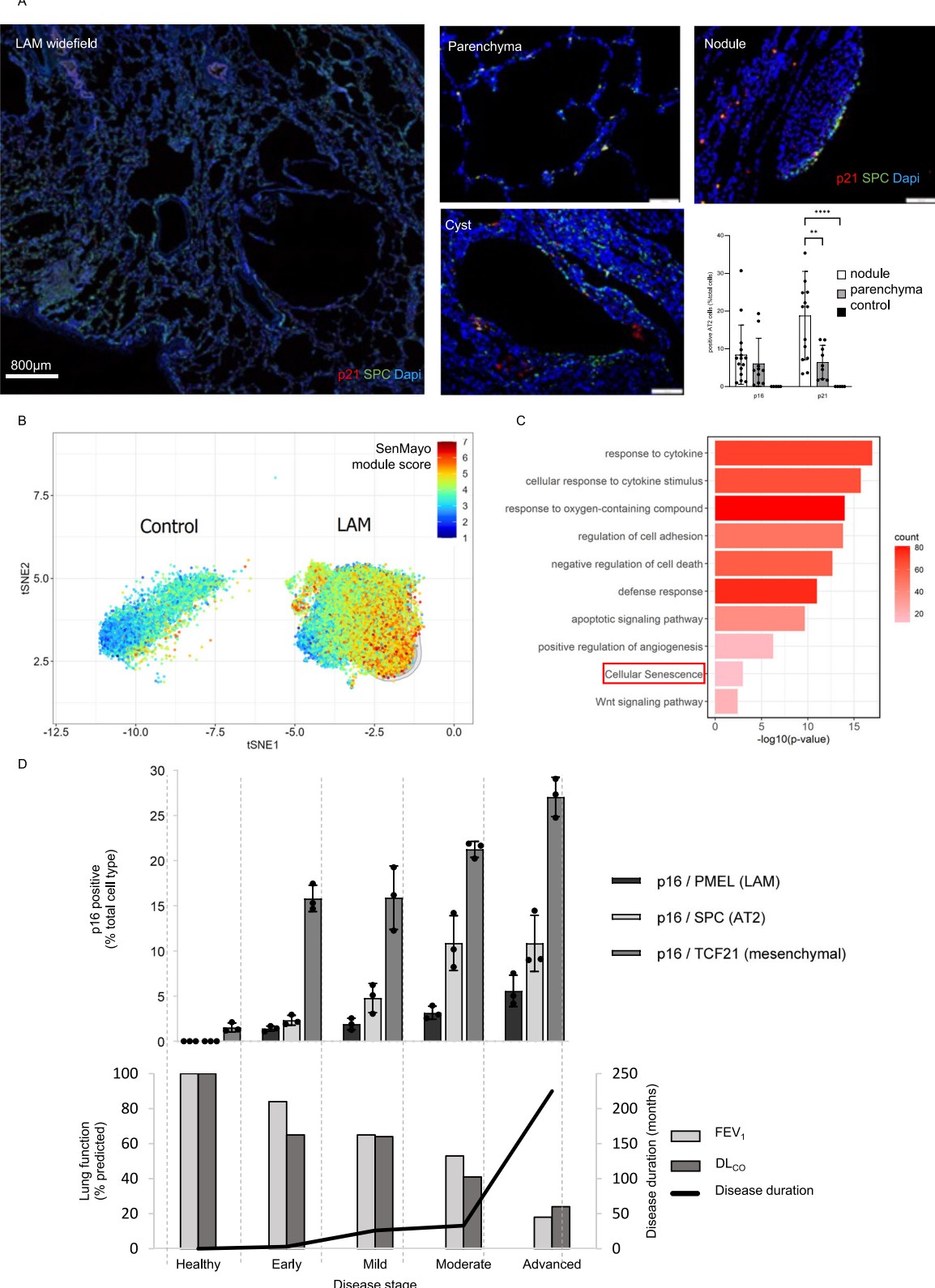

untreated with an mTOR inhibitor and five women receiving rapamycin for LAM. There was a greater than four-fold elevation of serum IL-6 in women with LAM (p = 0.0001) and was lower in persons treated with rapamycin (p < 0.05). Stratifying patients according to their lung function, showed an inverse correlation between serum IL-6 level and $FEV_1$ tertiles (p < 0.0001. Fig. 8F). Examining six-minute walk test distance as a functional measure of patient performance in 56 patients,

greater serum IL-6 values were seen in those with the lowest walk distance (r = −0.358 (95% C.I. −0.568 to −0.105), p = 0.007. Figure 8G). Serum IL-6 was also associated with lower post exercise saturation exercise (r = −0.281 (−0.498 to −0.0319), p = 0.028), gas transfer (r = −0.49 (−0.637 to −0.317), p = 1.06$^{-6}$) and rate of loss of gas transfer (r = −0.−317 (-0.521 to −0.0796), p = 0.01) but not with serum VEGF-D (Fig. 8H, and supplementary Fig. 14).

**Fig. 3 | AT2 cells are senescent in LAM. A** Representative widefield and close up images of dual-label immunohistochemical staining of p21 and the AT2 cell marker surfactant protein C (SPC) around LAM nodules, in the cyst walls and in more normal areas of lung parenchyma. Similar findings were observed with p16 (Supplementary Fig. 7). Graph shows co-expression of SPC and senescence markers p16 and p21 in LAM ($n \geq 9$, ≥3 replicates) and control ($n = 5$, ≥3 replicates). Data are presented as the mean (SD) percentage of positive cells relative to the total cells in regions of interest adjacent to LAM nodules or in control parenchyma. statistical analysis was performed using two-way ANOVA. Exact $p$-values are provided in the Source Data file (**$p < 0.01$, ****$p < 0.0001$). **B** Expression of the SenMayo gene set in control and LAM AT2 cells from LAM atlas[19]. The coloured bar represents the module score of the SenMayo gene set in AT2 cells, computed as the sum of all SenMayo gene unique molecular identifiers expressed in AT2 cells, divided by the sum of all unique molecular identifiers expressed in AT2 cells. **C** Differentially expressed pathways in LAM compared with healthy control AT2 cells. DEGs were identified using Wilcoxon Rank-Sum Test (two-sided test) with p value < 0.05 and fold change >1.5 without multiple comparison correction. An FDR of <10% was applied for pathway and gene set enrichment analysis. **D** Evolution of senescence-related changes in LAM lung tissue. The lower panel shows the disease stage in LAM patients ($n = 12$, ≥ 4 replicates) and control lung ($n = 3$, 3 replicates). Patients were stratified by lung function deficit for forced expiratory volume in 1 second ($FEV_1$) and lung diffusion of carbon monoxide ($DL_{CO}$). Normal (assumed 100% predicted), early (mean $FEV_1$ 84 [18]%), mild (65 [14]%), moderate (53 [30]%), and advanced disease (18%). The corresponding upper panel shows co-immunostaining with p16, SPC, TCF21 or PMEL to identify senescent AT2, LAFs and LAM/mesenchymal cells, respectively. Graph shows mean ± SD of the percentage of each p16-positive population relative to the total respective lineage. Source data and exact $p$-values are provided as a Source Data file.

## Discussion

We have shown that senescent cells accumulate in LAM nodules and lung parenchyma in a *TSC*2 / mTOR dependent manner. Our data suggest a sequence of events in which LAM cell mTOR dysregulation induces IL-6 secretion which promotes AT2 cell senescence to impair the alveolar epithelial repair response. As the disease progresses and LAFs accumulate in LAM nodules[7], these in turn become senescent, amplifying AT2 cell senescence and inducing senescence in a proportion of LAM cells (Fig. 9). Our findings point to a mechanism whereby senescence inhibits epithelial repair and increases the effect of lung injury in LAM and that inhibiting IL-6 signalling in parallel with mTOR inhibition may reduce lung damage in LAM.

Consistent with this idea, RNA sequencing showed increased activity in cytokine response pathways, likely attributable to SASP signalling in addition to injury as shown by increased apoptosis in lung parenchyma.

scRNAseq from LAM lungs showed that the SenMayo gene panel[25], a set of 125 genes which identify senescent cells in multiple settings was increased in LAM AT2 cells. Utilising a murine transgenic mesenchymal specific *Tsc2* deletion we observed that rapamycin could suppress AT2 cell senescence when instituted early in development. scRNAseq from a single patient treated with an mTOR inhibitor showed that AT2 cell related senescent gene expression was reduced by mTOR inhibitor therapy but once established, in a further single mTOR inhibitor treated patient senescent cells persisted in the lungs despite mTOR inhibition. Data using a *TSC*2 null homograft mouse also confirmed the mTOR dependent accumulation of senescent AT2 cells but no significant reduction in senescence related proteins with mTOR inhibition once the disease was established. Whilst the availability of mTOR inhibitor treated patient lung limits the strength of the conclusions in humans and histological markers of LAM cells and LAFs are not completely sensitive nor specific, meaning quantification of LAM nodule components lacks complete precision, it is clear that mTOR dysregulation induces AT2 cell senescence in animals and using multiple closely phenotyped patient samples, in humans senescent AT2 cells and LAFs followed by senescent LAM cells accumulate with disease progression.

Senescence is a process of irreversible cell cycle arrest, thought to have evolved to suppress tumorigenesis in cells prone to DNA damage and can be induced by high levels of cell division, injurious processes including ROS, DNA damage or oncogene activation including that resulting in mTOR dysregulation[26]. We had considered that *TSC*2 loss/ mTOR dysregulation would initially induce senescence in LAM cells. Whilst scRNAseq data showed a doubling of senescence related genes in LAM^CORE cells, our findings suggest that mTOR driven LAM cell IL-6 secretion is likely to be the initiating event, with fully senescent p16 / p21 expressing LAM cells a feature of late-stage disease only. Importantly the single cell analysis was made predominantly on whole lung explants from patients with the most severe disease in contrast to the histological samples which were selected to study the full spectrum of disease evolution including early disease. Thus the LAM scRNAseq data from transplant tissue, was consistent with our findings in which senescent LAM cells were only seen in patients with advanced disease. In vitro, and consistent with the work of Bernadelli and colleagues[27], *TSC*2 null cells express SAβgal, p16, p21. Additionally, we show that *TSC*2 null cells express IL-6 at higher levels than their *TSC*2 addback counterparts and in co-cultures LAM derived 621-101 cells and 621-101/ LAF cultures increase expression of SAβgal in LAFs and alveolar epithelial cells to a greater extent than *TSC*2 addback 621-103 cells and 621-103/LAF co-cultures. We also observe that LAFs express senescence markers and can also increase SAβgal expression in LAM derived 621-101 cells which was not the case in the *TSC*2 addback 621-103 cells, suggesting that the mTOR dependent SASP can induce paracrine senescence within LAM nodules. Importantly, the LAM nodule evolves over time with fibroblasts increasingly recruited by LAM cells[3,7], suggesting the pattern of cell senescence is dependent upon both mTOR dysregulation and the evolution of the disease (Fig. 9).

Senescence is a feature of other lung diseases, particularly those affecting older people where replicative senescence and consequent p53 activation are important in addition to ROS in COPD, and telomere dysfunction in idiopathic pulmonary fibrosis (IPF)[15,28]. In COPD, activation of mTOR signalling is associated with induction of MicroRNA-34a suppressing sirtuin-1 to promote p53-induced senescence and an NF-κB, mediated SASP response[15,29], with rapamycin both inhibiting senescence and the SASP[30]. Telomere dysfunction, aging and other processes induce myofibroblast senescence in IPF, and in adjacent cells via SASP proteins[31]. People with LAM are younger than those affected by COPD and IPF and also less likely to smoke[32]. Our findings suggest that cell-cell interactions between mTOR dysregulated LAM cells and LAFs within LAM nodules drive senescence, possibly less dependent upon aging, ROS production or telomere dysfunction seen in other lung diseases.

We used several independent methods to understand how cell-cell communication within the LAM lung leads to AT2 cell senescence which could adversely affect the response to lung injury. RNAseq of lung parenchymal cells highlighted activation of cytokine response genes, consistent with SASP signalling, defence against injury, apoptosis and senescence reflecting the LAM related parenchymal damage and the repair response to that injury. We focused upon the AT2 cell, the stem cell responsible for lung repair and prone to senescence[33]. Senescent AT2 cells were not detected in healthy lungs but present in the AT2 cells surrounding LAM nodules[24], in cyst walls and in apparently more normal appearing areas of LAM lung.

LAM spheroids (3D LAM cell/LAF co-cultures) induced p16 and p21 in human AT2 cell organoids, dependent on mTOR activation, potentially consistent with SASP factors affecting wound repair processes and thus contributing to lung cyst formation. Whilst it is difficult to relate senescence directly to wound repair in these models, the induction of these cell cycle inhibitors is likely to account for the altered AT2 organoid growth in the presence of both *TSC*2 null and

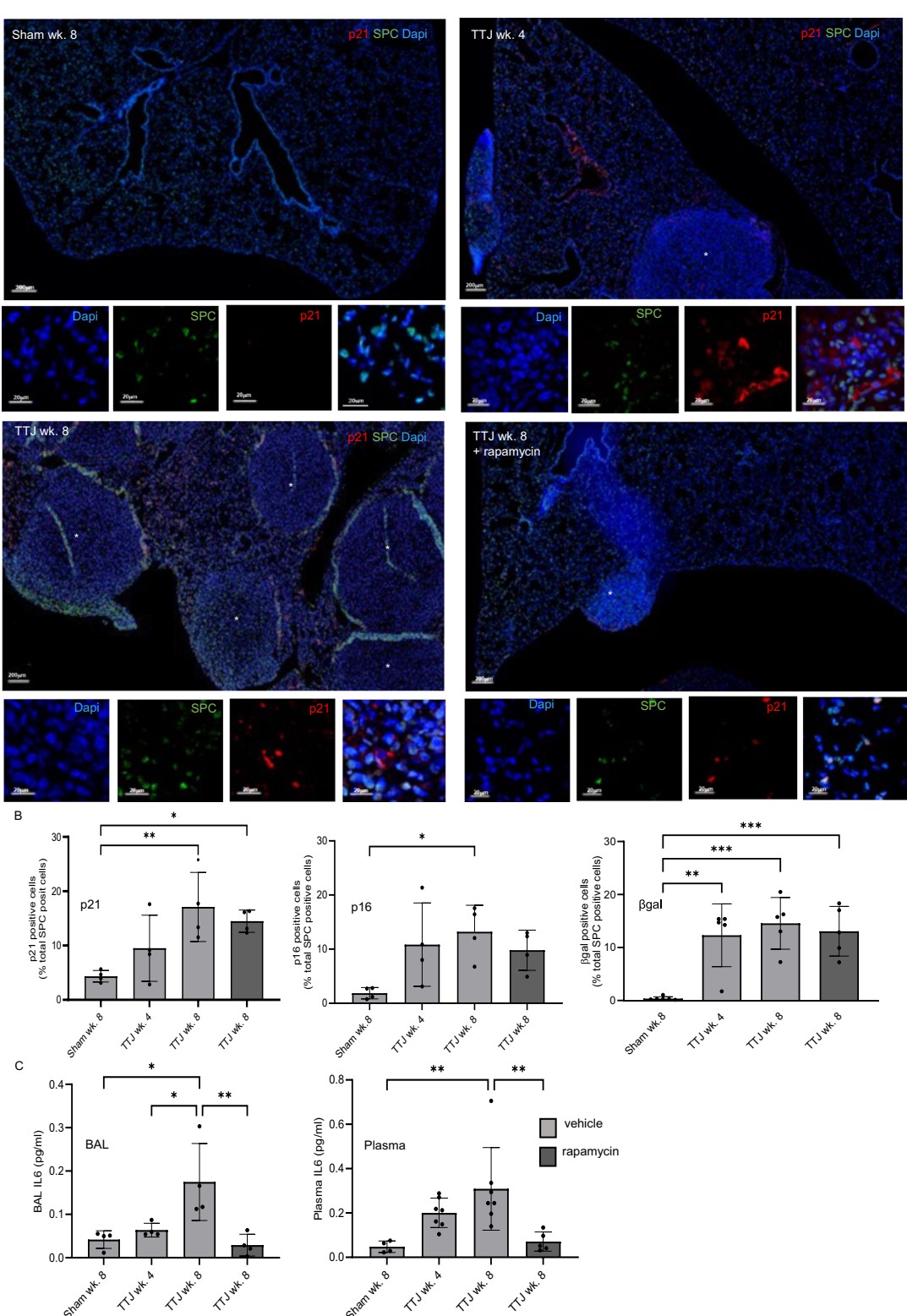

**Fig. 4 | Senescence and IL-6 secretion increase with time and disease burden.**
Time course of senescent cell and SASP generation was analysed in an animal model. TTJ cells or saline (sham) were injected into the tail veins of albino C57BL/6 mice, animals were treated with rapamycin or vehicle two weeks after cell injection, tissue, bronchoalveolar lavage fluid (BAL) and serum were harvested at 4 and 8 weeks after cell injection. **A** Senescent AT2 cells were quantified by dual immunostaining with SPC and p21, p16 or beta-galactosidase (βgal), representative images for p21 are shown. * highlights tumour nodules. **B** Quantification of mean (SD) dual expressing SPC / p21, p16 or βgal positive cells at different time points and the effect of the mTOR inhibitor rapamycin ($n = 3$ / group). Data analysed by one-way ANOVA. Source data and exact p-values are provided as a Source Data file. *$p < 0.05$, **$p < 0.01$, ***$p < 0.001$, ****$p < 0.0001$. **C** Mean (SD) IL-6 protein in plasma ($n \geq 4$, duplicate) and bronchoalveolar lavage (BAL) ($n \geq 4$, duplicate), data analysed by one-way ANOVA. Source data are and exact p-values provided as a Source Data file *$p < 0.05$, **$p < 0.01$.

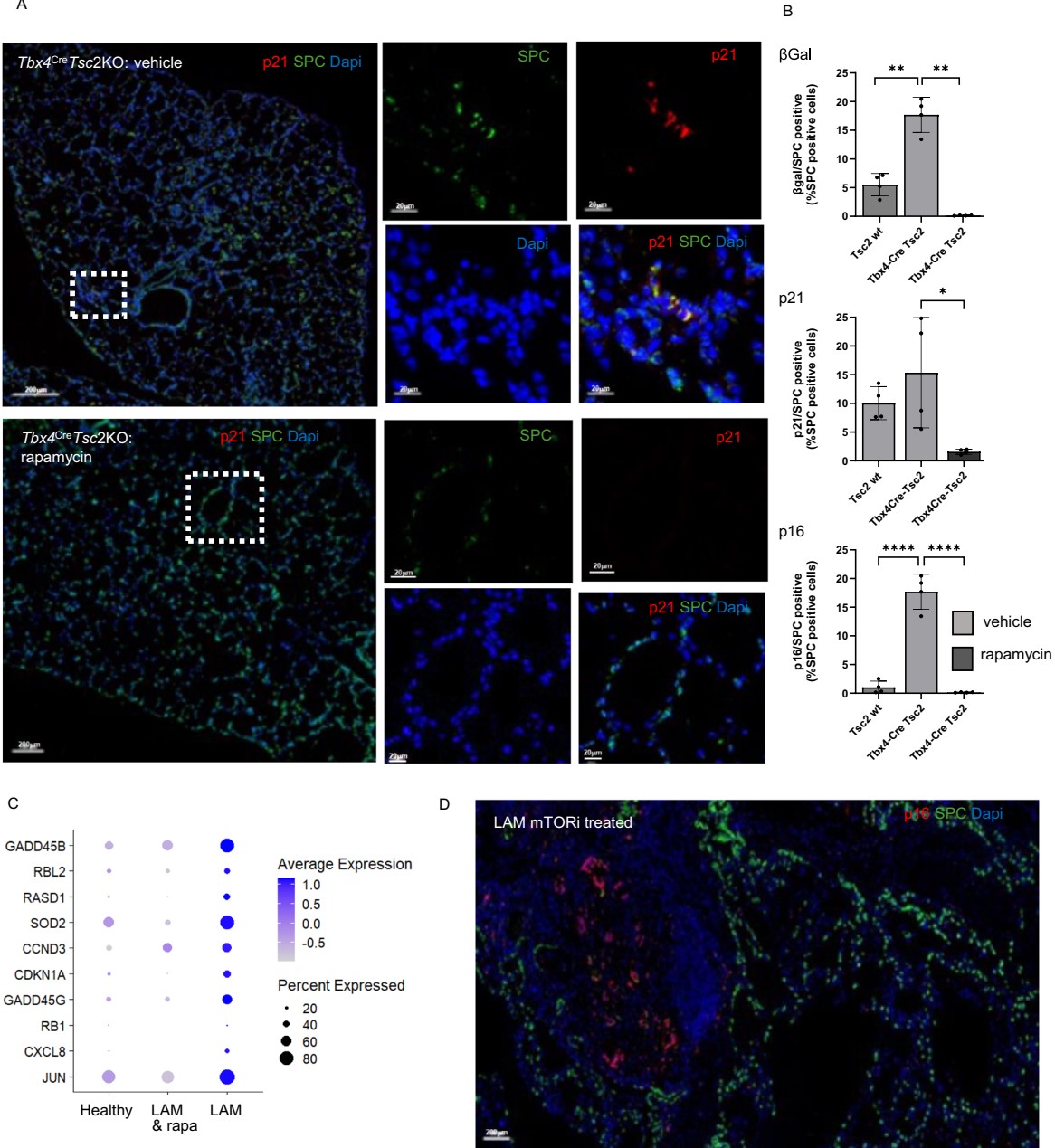

**Fig. 5 | AT2 cell senescence is mTOR dependent.** *Tsc2* was deleted in mesenchymal cells using a *Tbx4*[Cre]*Tsc*2KO mouse[10]. Animals were treated with rapamycin from birth and lung tissue harvested at 20 weeks of age. **A** Representative images of *Tbx4*[Cre]*Tsc*2KO animals with and without rapamycin. **B** Mean (SD) SAβgal/SPC, p21/SPC and p16/SPC dual positive AT2 cells were quantified in four animals / group including *Tsc2* wild type control animals analysed by one way ANOVA (*n* = 4). Source data and exact *p*-values are provided as a Source Data file. *$p < 0.05$

**$p < 0.01$, ****$p < 0.0001$. C Expression of a panel of senescence associated genes from single cell RNA sequencing from LAM and healthy control AT2 cells. Single cell RNA sequencing from a single lung from a patient with LAM treated with the mTOR inhibitor rapamycin (LAM & rapa) showing suppression of senescence associated genes. **D** Dual immunostaining with SPC and p16 in lung tissue from a single lung transplant in a patient treated with the mTOR inhibitor Everolimus.

*TSC*2 addback LAM spheroids. Whilst these in vitro changes over 14 days were not significant, it is likely that the AT2 cell senescence we observed in human LAM represents a similar process, which could enhance lung cyst formation over the disease course. Upstream regulator analysis highlighted the IL-6 receptor as a potential effector of AT2 cell senescence. We showed that IL-6 secretion was downstream of mTOR in 621-101 cells, in a mouse model of LAM, and also overexpressed in human LAM where it was related to lung function, exercise performance and importantly disease activity as assessed by rate of lung function loss. IL-6 inhibited wound repair and induced AT2 cell SAβgal and p16 expression in vitro. Interestingly, although LAM

spheroids did not alter scratch wound repair, treatment with the IL6R inhibitor Tocilizumab strongly enhanced repair suggesting IL6R signalling inhibits wound repair but other features of the LAM spheroid secretome have competing effects on these processes. The elevation of serum IL-6 in women with LAM[34], its relationship to the extent of lung involvement and patient performance, makes IL-6 signalling a potential target for therapy. Both rapamycin and Tocilizumab suppressed the accumulation of AT2 cell p16 in a model containing LAM cells and LAFs and the combination of the two drugs reduced p16 further. The production of IL-6 by LAFs that do not have *TSC* loss and the lack of correlation between serum levels of IL-6 and VEGF-D,

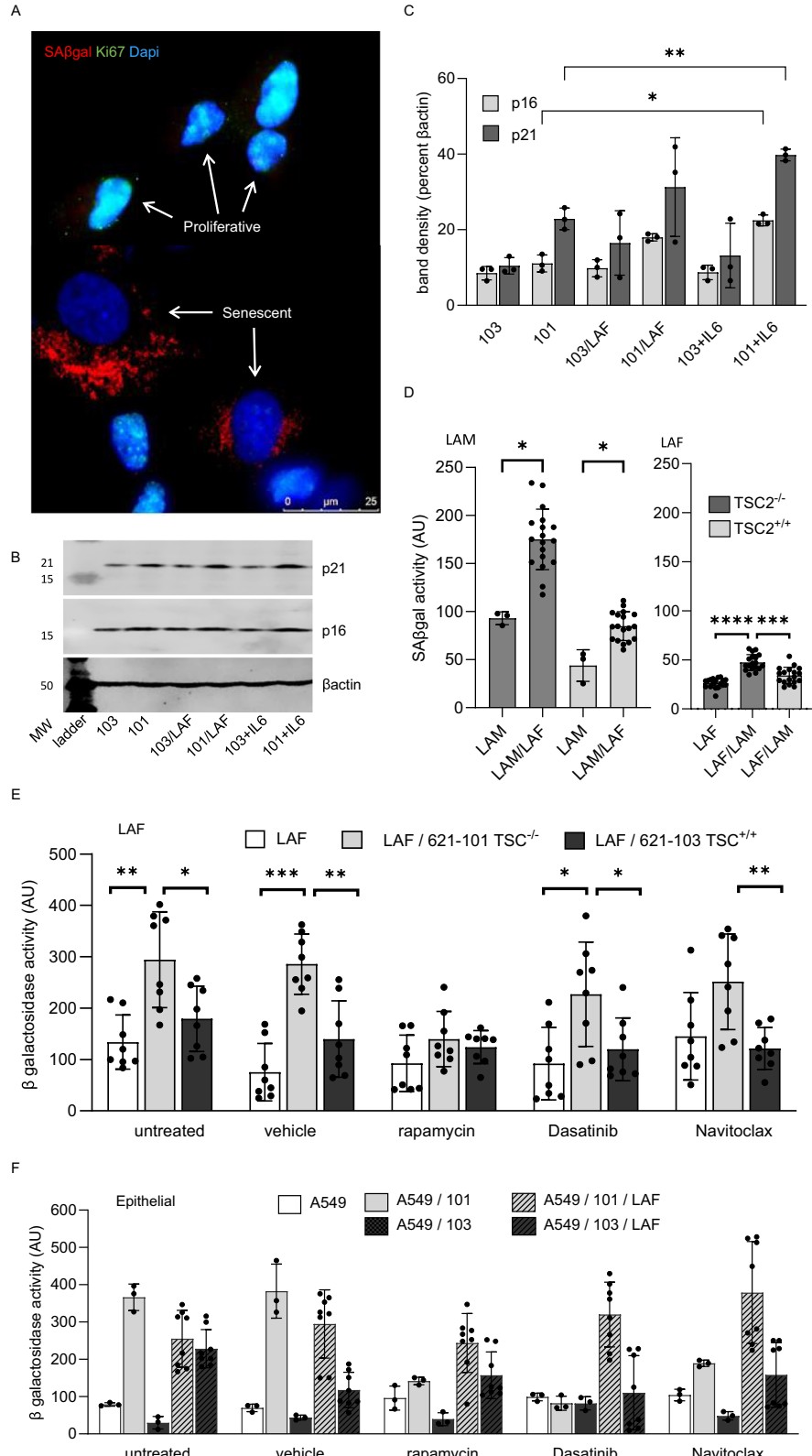

thought to be a readout of pathological mTOR signalling in women with LAM, suggest that IL-6 secretion may be mTOR dependent in LAM cells but not LAFs. Although effective, patients may still lose lung function whilst treated with rapamycin, presumed to be due to mTOR independent pathways[35]. While the absence of an animal model containing all features of the disease means our work used a LAM cell / LAF long term co-culture to study these interactions, it is at least likely that

combined mTOR / IL-6 at antagonism may have an additional benefit over rapamycin alone, particularly in more advanced disease.

Here, we show that senescent cells accumulate in both LAM nodules and the AT2 cell population and these processes including IL-6 secretion are mTOR dependent. Our findings suggest that AT2 cell senescence induced by LAM cell derived IL-6 occurs early in the disease course, with the induction of senescence in LAFs amplifying the

**Fig. 6 | Cell-cell interactions induce senescence dependent on mTOR dysre-gulation. A** Dual-immunocytochemical staining for senescence associated beta galactosidase (SAβgal) and the proliferation marker Ki67 in 621-101 cells main-tained in low serum cultures over 14 days. Proliferative cells show nuclear Ki67 staining and sparse SAβgal positive lysosomes, whereas senescent cells are large, Ki67 negative, with high levels of SAβgal containing lysosomes. This experiment was repeated 4 times independently with similar results. **B** Western blot of p21, p16 and loading control, beta actin (βactin) in 621-101 and 621-103 cells (*TSC2* addback control), untreated, co-cultured with LAM associated fibroblasts (LAF) or treated with IL-6. MW is molecular weight in kilo Daltons. **C** Mean (SD) densito-metry of western blots of three independent experiments described in panel B (*n* = 3). Data analysed by 2-way ANOVA. *$p < 0.05$ (**D**) Interactions between LAM derived 621 cells (LAM), LAM associated fibroblasts (LAF) and LAM/LAF co-cultures assayed after 14 days (*n* = 4, 3 replicates). Graphs show mean (SD) percentage of

LAM cells (left panel) or LAFs (right panel) expressing SAβgal when interacted in culture. *TSC2*$^{+/+}$ are control, addback 621 cells used to show the effect of mTOR dysregulation on the induction of senescence. Data analysed by multiple unpaired t-tests with a false discovery rate of 1% for multiple comparisons. *$p < 0.05$ and ****$p < 0.0001$. **E** Mean (SD) senescence associated beta galactosidase activity in LAFs (*n* = 4 duplicates) when cultured with 621 cells and the effect of rapamycin and the senolytic drugs Dasatinb and Navitoclax. Data are analysed by multiple unpaired two tailed t tests with a false discovery rate of 1% for multiple compar-isons. *$p < 0.05$ and **$p < 0.01$, ***$p < 0.001$. **F** Mean (SD) senescence associated beta galactosidase activity in A549 epithelial cells when cultured with *TSC2*$^{-/-}$ 621-101 cells, *TSC2*$^{+/+}$ 621-103 cells and LAFs (n = 4 duplicates) in combination and the effect of rapamycin, Dasatinb and Navitoclax. Panels F analysed by 2-way ANOVA. All source data and exact *p*-values are provided as a Source Data file. *$p < 0.05$, **$p < 0.01$, ****$p < 0.0001$.

SASP, potentially altering epithelial cell growth and repair (Fig. 9). Whilst it is difficult to definitively link senescence and lung repair, these processes are related and our findings highlight that targeting SASP factors including IL-6 in parallel with mTOR inhibition, could potentially reduce lung damage in LAM.

## Methods
Full methodological details and antibody details are provided in the online supplement.

### Patient samples
Serum samples and lung tissue were obtained from patients receiving care at the UK LAM Centre between 2011 and 2022. All patients had LAM as defined by American Thoracic Society / Japanese Respiratory Society criteria[36]. Tissue samples were taken for clinical care with clinical data captured from medical notes. Lung function and VEGF-D were recorded at baseline and lung function repeated at subsequent visits. Change in lung function was calculated as the slope of a regression line of all values of FEV$_1$ (ΔFEV$_1$) or DL$_{CO}$ (ΔDL$_{CO}$).

### Cell, LAM spheroid and alveolar organoid culture
621-101, control *TSC2*-add-back 621-103 cells, TTJ cells derived from renal tumours of *TSC2*$^{+/-}$ C57BL/6 mice primary LAFs and LAM spher-oids have been described previously[6]. AT2 cells were derived from human foetal lung tissue obtained under ethical approval (refer-ence18:/LO/0822; Project 200591; (www.hdbr.org)[37] and cultured in organoids as described[38,39].

### Laser capture microdissection, library Preparation and RNA Sequencing
Tissue sections were mounted on PEN Membrane Glass Slides (Thermo Fisher scientific LCM0522) and LCM performed on a Zeiss AxioImager Z1 (Zeiss, Thornwood, NY), using αSMA and PNL2 to identify LAM nodules. RNA was extracted using the RNeasy DSP FFPE Kit (QIAGEN, ID: 73604) and RQ1 RNase-Free DNase (Promega, M6101) with clean-up performed using Monarch RNA Cleanup Kit (Biolabs, New England, T2030L). RNA was quantified using Ribogreen QuanT-iT kit and quality checked using a high sensitivity RNA Tape on a 4200 TapeStation. Samples with DV200 score (RNA greater than 200 bp in length) >35% were analysed. Libraries were prepared using Lexogen QuantSeq 3' mRNA-Seq Library Prep Kit FWD and quantified using a PicoGreen Quant-iT kit and sized with D1000 tape. Pooled libraries were quality checked using the Agilent TapeStation DNA 1000 tape and DNA HS Qubit and loaded on a NovaSeq 6000 with an SP 100 cycle flowcell.

### Bulk RNA-seq analysis
RNA-seq data alignment, QC and normalisation were performed using Partek Flow (v7. Partek Inc., St. Louis, MO, USA). Human genome hg38 was used as reference. Gene counts were normalised as the ratio of the read count to the geometric mean across all samples and subjected to

differential expression analysis[40]. DEGs were identified using DESeq2 in Partek with a *p* value < 0.05 and absolute fold change >1.5.

### Single cell RNA sequencing analysis
scRNAseq data were retrieved from the LAM Cell Atlas with scRNAseq data QC, pre-filtering, batch correction and data integration per-formed as previously described[41]. 65,287 cells from 11 LAM lungs (13 biological replicates) and 43,128 cells from 8 control female lungs[10,42] were included in integrative analysis using Seurat 3[43]. Unbiased cell clustering was performed using the Leiden algorithm[44]. Cell clusters and cell type annotation were performed based on known markers and signature genes from our previous LAM and normal lung single cell studies[41,45–47]. Integrated single cell analysis identified an AT2 cluster consisting of 12,291 LAM AT2 cell and 17,378 control AT2 cells. The Wilcoxon rank-sum test was used to compare the gene expression levels between AT2 cells in LAM and control. Fold change was calcu-lated as the ratio of the average expression level in the AT2 populations in LAM lungs and in normal control lungs. Functional enrichment analysis was performed using ToppGene suite[48]. CellChat[49] analysis was applied on the scRNA-seq data to decipher LAM$^{CORE}$ and LAM AT2 cell communication patterns based on cell type selective expression of ligands secreted from LAM$^{CORE}$ cells and receptors expressed on the surface of LAM AT2 cells. Ingenuity Pathway Analysis was used to predict activated upstream regulators using the differentially expres-sed senescence genes in LAM AT2 cells as input genes.

### Immunohistochemistry
Paraffin-embedded tissues were dewaxed, rehydrated and antigen retrieval carried out in heated sodium citrate or Tris-EDTA buffer peroxidase activity quenched. Detection was performed using ImmPACT 3,3'diaminobenzidine (DAB) peroxidase substrate (Vector Laboratories, SK-4105).

### Immunofluorescence
Organoids were collected after centrifugation, fixed with either par-aformaldehyde (PFA) or ethanol based on the primary antibody pro-tocol, permeabilized and blocked with BSA. Primary antibodies were incubated at 4 °C overnight and secondary antibodies incubated for 1 hour room temperature and counterstained with DAPI.

### Senescence associated β-galactosidase activity
SAβgal was measured in cell cultures using a plate-based fluorescent assay (Cell Signalling, 23833). In tissue SAβgal activity at pH 6 was determined using either Senescence β-Galactosidase Staining Kit (cell signalling, 9860) or the CellEvent Senescence Green Detection Kit (Invitrogen, C10850) according to the conditions.

### Animal models
In the homograft system 10$^6$ TTJ cells or saline control were injected into the tail vein of female albino C57BL/6 (B6N-Tyrc-Brd/BrdCrCrl)

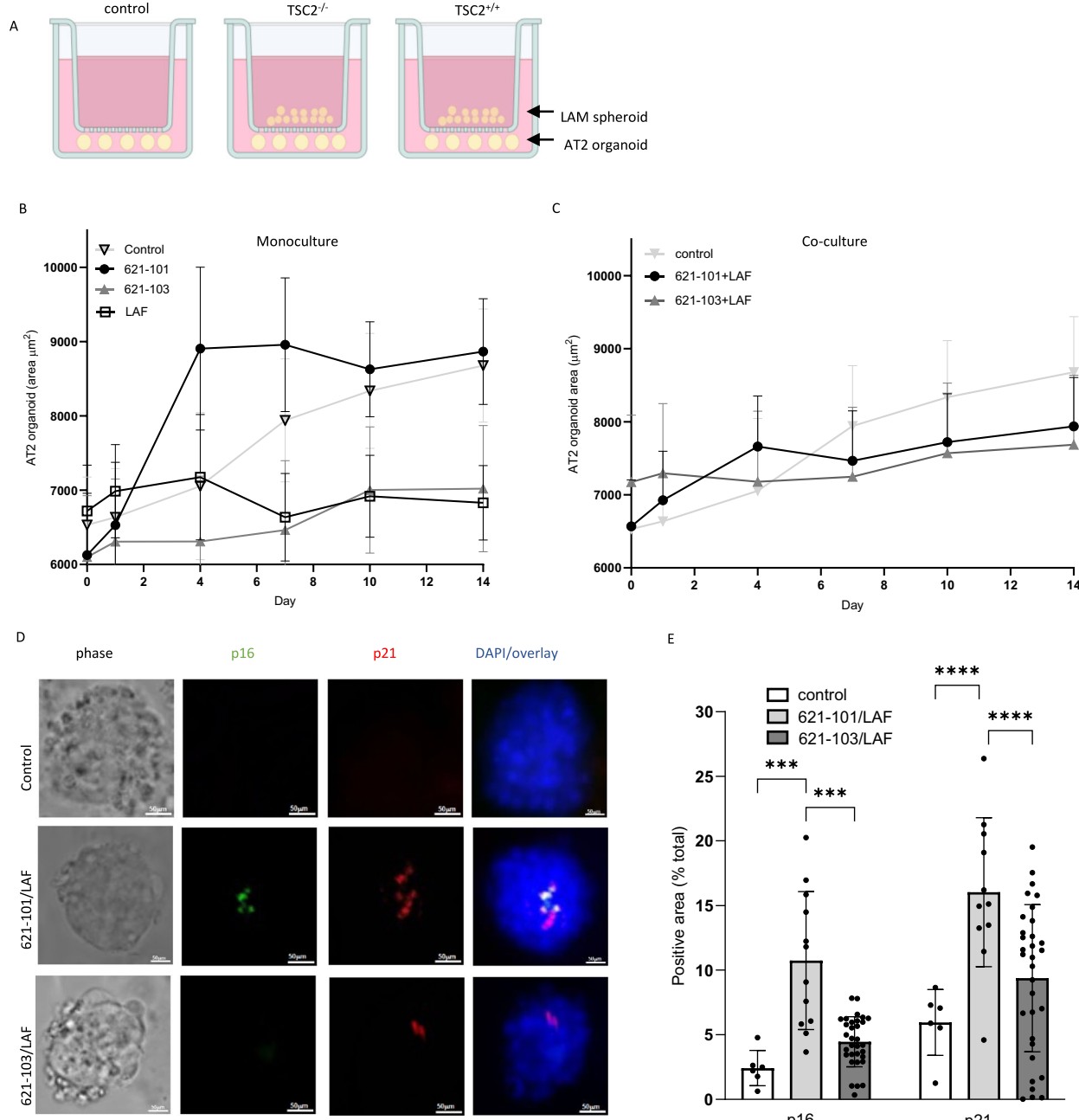

**Fig. 7 | LAM nodules induce AT2 cell senescence in vitro. A** Experimental setup showing LAM spheroids: 3D cocultures of *TSC2⁻/⁻* or *TSC2⁺/⁺* LAM derived 621 cells and LAM associated fibroblasts (LAF) cultured in transwells with AT2 cell organoids over 14 days Created in BioRender. Clements, D. (https://BioRender.com/qrl1vu9). **B** Mean (SEM) AT2 cell organoid area over 14 days co cultured with *TSC2⁻/⁻* 621-101 cells, *TSC2⁺/⁺* 621-103 cells or LAFs (*n* = 5 LAF donors, ≥5 replicates). **C** Mean (SEM) AT2 cell organoid area over 14 days co cultured with *TSC2⁻/⁻* 621-101/LAF or *TSC2⁺/⁺*

621-103/LAF co-cultures. **D** Induction of the senescence markers p16 and p21 in AT2 cell organoids over 14 days (*n* = 5 LAF donors, at least triplicate). **E** Quantification of mean (SD) p16 and p21 protein in AT2 cell organoids co-cultured with *TSC2⁻/⁻* and *TSC2⁺/⁺* (*n* = 4 LAF donors, at least triplicate). LAM spheroids analysed by 2-way ANOVA. Source data and exact *p*-values are provided as a Source Data file. \**p* < 0.05 and \*\**p* < 0.01, \*\*\**p* < 0.001, \*\*\*\**p* < 0.0001.

conducted under Home Office project Personal Project Licence P435A9CF8 as described[6] n ≥ 4 in each group). The transgenic model, using a conditional *Tsc2* knock-out gene in lung mesenchyme, on aTbx4LME_Cre background[50], using female mice (*n* ≥ 4 per group) was performed as described[10]. Animals were treated with rapamycin (4 mg/Kg, i.p.) or vehicle from birth and lung tissue harvested at 20 weeks. Animals were euthanised by anaesthetic overdose followed by femoral artery exsanguination.

## Western blotting

Nuclear and cytoplasmic protein fractions were separated as described[51]. Blots were probed with primary antibodies overnight and secondary antibodies for one hour room temperature. Primary antibody p16INK4A Rabbit Polyclonal antibody (1:100, Proteintech No. 10883-1-AP), p21 Waf1/Cip1 (12D1) Rabbit mAb (1:100 Cell Signalling Technology No: 2947) and β-actin Rabbit mAb (Cell signalling, 4970) was used as a loading control. Protein bands were

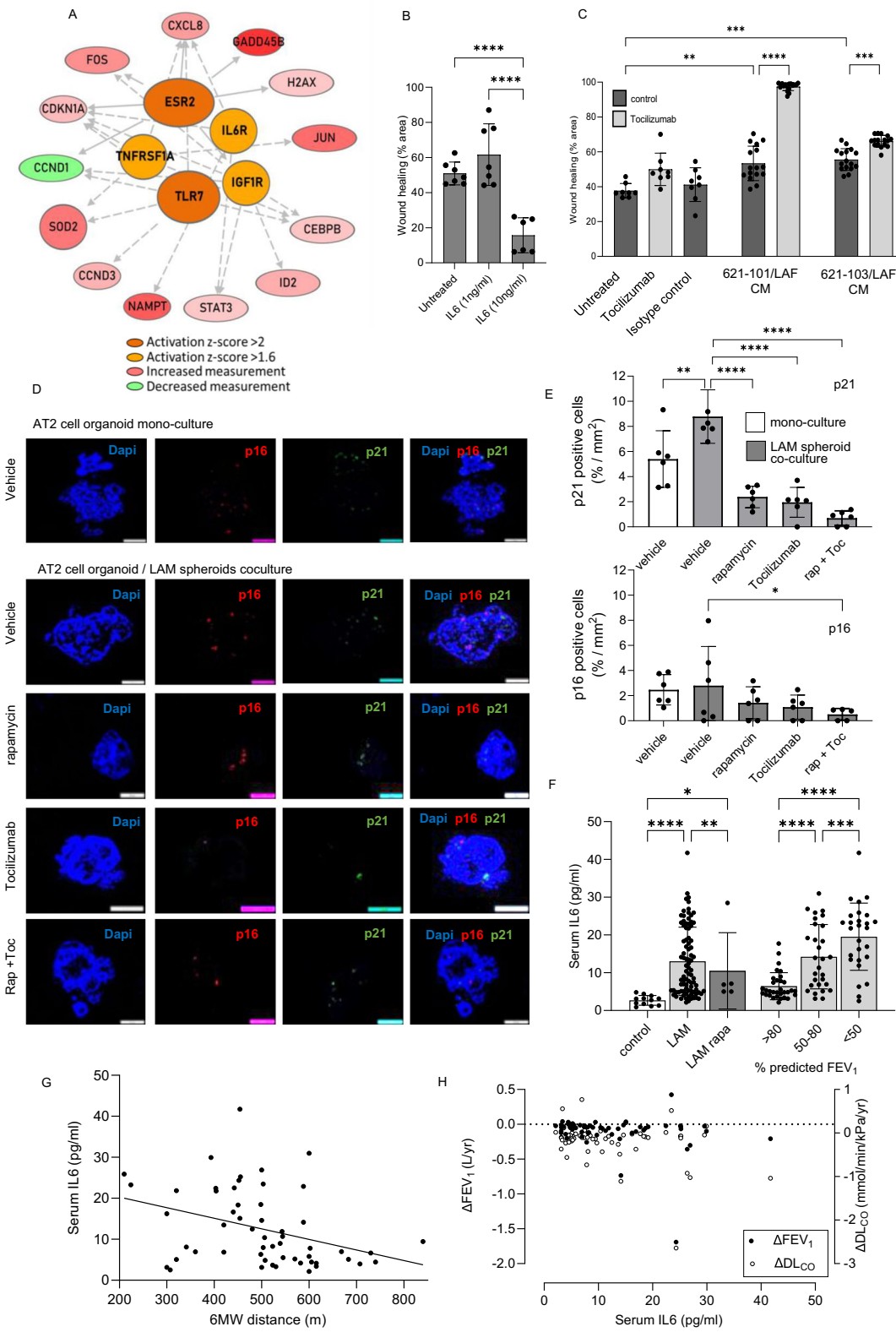

visualised using Clarity Max Western ECL Substrate (BIO-RAD 1705062).

**IL-6 ELISA**

IL-6 in conditioned media or serum was measured using a Human IL-6 DuoSet ELISA (R and D, DY206) according to the manufacturer's protocol.

**Statistics & reproducibility**

Data were analysed by Mann-Whitney test, two tailed t tests and two-way ANOVA using the Dunnett multiple comparisons method as appropriate using Graph Pad Prism 10 (GraphPad Software Boston, MA). The correlation of p16 and p12 positive cells with future lung function change, adjusted for baseline lung function, was analysed in PRISM. $P < 0.05$ was considered as indicating significance. No

**Fig. 8 | IL-6 signalling induced by LAM nodules affects wound repair in vitro.**
**A** Upstream regulator analysis of AT2 cell gene induction predicted activation of oestrogen receptor β (ESR2), the interleukin 6 receptor (IL6R), toll like receptor 7 (TLR7), TNF receptor superfamily 1 A (TNFSF1A) and insulin dependent growth factor receptor 1 (IGF1R). **B** The effect of IL-6 on scratch wound repair in A549 cells. Graphs show mean (SD) percent of wound area healed over 24 h ($n = 8$, triplicates) analysed by one-way ANOVA. **C** The effect of IL-6 inhibition using Tocilizumab in scratch assay in A549 cells incubated with conditioned medium (CM) from 3D co-cultures of $TSC2^{-/-}$ 621-101 or $TSC2^{+/+}$ 621-103, and LAF. Data are mean (SD) ($n = 8$) analysed using multiple unpaired two-tailed t-tests with 1% FDR correction. Panel C-D Source data and exact p-values are provided (*$p < 0.05$, **$p < 0.01$, ***$p < 0.001$,). **D, E** Expression of senescence markers p16 and p21 was assessed in AT2 organoids (14-days) culture, either in monoculture or co-culture with 621-101-LAF spheroids, in the presence of rapamycin

(10 ng/ml), Tocilizumab (30 ng/ml), or combination therapy. Representative images (D) and quantification (E) show increased expression of senescence markers in LAM spheroid co-culture, attenuated by Tocilizumab and further reduced by combined treatment. Data represent ≥5 organoids in triplicate, analysed by one-way ANOVA. *$p < 0.05$ and **$p < 0.01$, ***$p < 0.001$, ****$p < 0.0001$. **F** Serum IL-6 was measured in healthy women ($n = 19$) and in women with LAM ($n = 65$), either untreated or treated with rapamycin. Stratification by lung function (FEV$_1$ % predictaed) at the time of sampling analysed by 2-way ANOVA. Source data and exact p-values are provided as a Source Data file. *$p < 0.05$ and **$p < 0.01$, ***$p < 0.001$, ****$p < 0.0001$. **G, H** Higher serum IL-6 was associated with reduced six-minute walk (6 MW) distance ($n = 56$, $p = 0.0067$) and with more rapid decline in FEV$_1$ and DL$_{CO}$ in LAM patients ($n = 88$). Linear regression analyses confirmed significant associations. Source data and exact p-values are provided in the source data file (**$p < 0.01$, ***$p < 0.001$, ****$p < 0.0001$).

## Table 1 | Predicted AT2 cell receptor activation from up-stream regulator analysis

| Upstream Regulator | Molecule Type | Activation (z-score) | Target Molecules in Dataset |
|---|---|---|---|
| *ESR2* Oestrogen receptor β | ligand-dependent nuclear receptor | 2.209 | *CCND1, CDKN1A, CXCL8, FOS, GADD45B, H2AX* |
| *TLR7* Toll-like receptor 7 | transmembrane receptor | 2.178 | *CCND1, CCND3, CDKN1A, CEBPB, CXCL8, SOD2* |
| *TNFRSF1A* TNF receptor superfamily member 1A | transmembrane receptor | 1.994 | *CDKN1A, CEBPB, CXCL8, JUN, SOD2* |
| *IL6R* Interleukin 6 receptor | transmembrane receptor | 1.982 | *CDKN1A, CXCL8, NAMPT, STAT3* |
| *CAV1* Caveolin 1 | transmembrane receptor | 1.911 | *CCND1, CDKN1A, FOS, ID1* |
| *IGF1R* Insulin dependent growth factor receptor 1 | transmembrane receptor | 1.698 | *CCND1, CDKN1A, CEBPB, FOS, ID2, STAT3* |
| *GPER1* G Protein-Coupled Oestrogen Receptor | G-protein coupled receptor | 1.679 | *CCND1, CDKN1A, FOS, JUN, RASD1* |

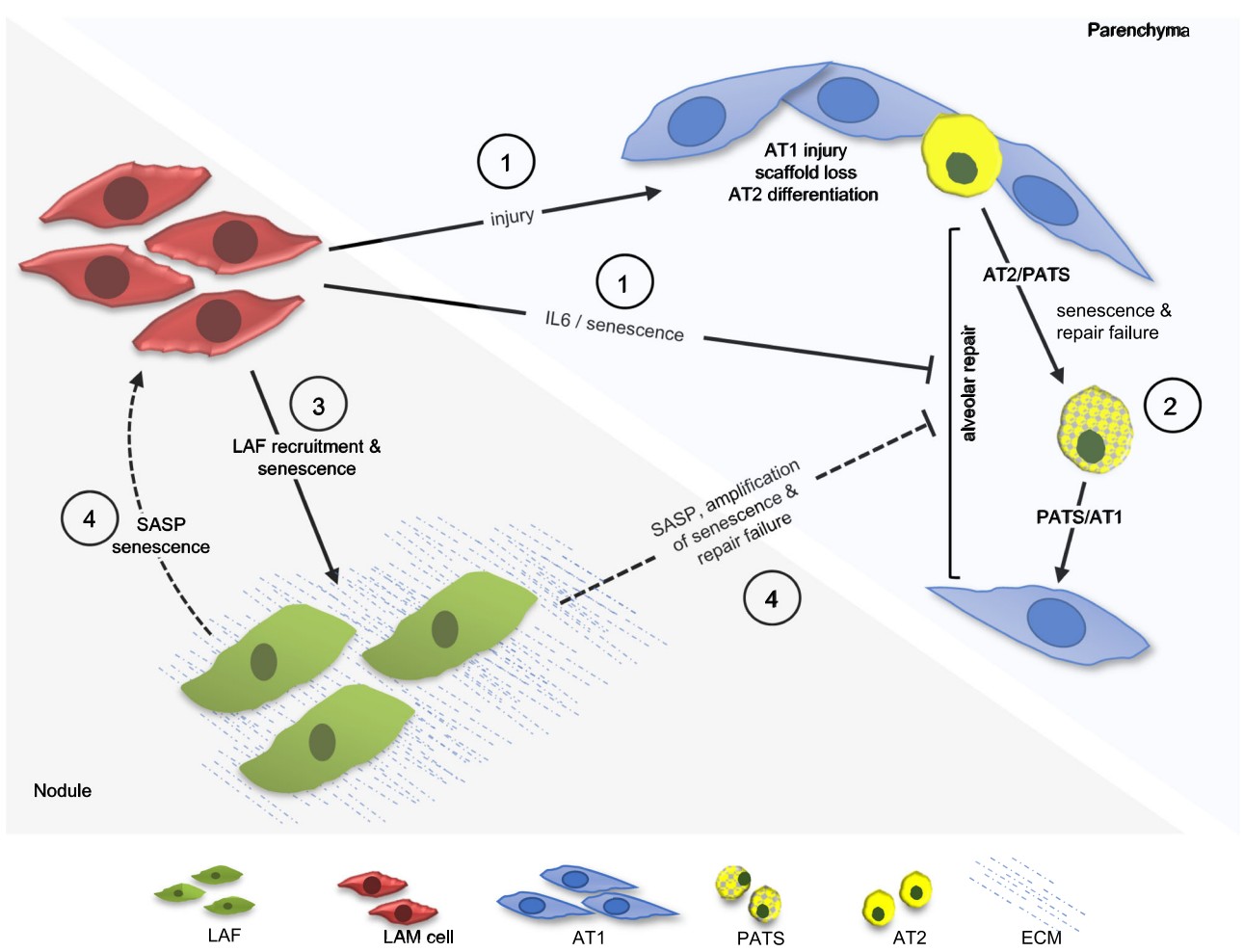

**Fig. 9 | Summary of cell-cell interactions leading to lung repair failure in LAM. 1**
LAM cells secrete proteases causing lung injury. Simultaneously, mTOR dysregulated LAM cells secrete IL-6 inducing AT2 cell senescence and impairing lung repair. **2** Lung injury induces alveolar type 2 cell (AT2) differentiation to alveolar type 1 cell (AT1) via intermediate prealveolar type-1 transitional cell state (PATS). **3** As the

disease progresses, LAM cells recruit LAM associated fibroblasts to LAM nodules and induce LAF senescence. **4** In later disease, senescent LAFs induce LAM cell senescence and amplify AT2 cell senescence increasing tissue damage, Created in BioRender. Clements, D. (https://BioRender.com/6bdk7z6).

statistical method was used to predetermine sample size due to the limited availability of fixed lung samples, meaning pilot experiments to determine statistical power could not be conducted. AT2 cell organoid experiments were performed on samples from three donors, with sample sizes based on a previous studies.

## Ethics statement

The use of patient data, serum and lung tissue was approved by the East Midlands Research Ethics Committee (reference 13/EM/0264) and all participants gave written informed consent.

## Reporting summary

Further information on research design is available in the Nature Portfolio Reporting Summary linked to this article.

## Data availability

LAM lung RNA sequencing data are available at GEO under primary accession number GSE265851. LAM lung single cell data reused here are available through the LAM cell atlas (https://research.cchmc.org/pbge/lunggens/LCA/LCA.html). Link anonymised clinical data for the single cell sequencing are available to academic groups through the corresponding author. All other data are available in the article and its Supplementary files or from the corresponding author upon request. Source data are provided with this paper.

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

## Acknowledgements
The study was funded by Medical Research Council grant (MR/T002042/1), LAM Action, Fundació La Marató De TV3 and the Nottingham NIHR Biomedical Research Centre to SRJ. ELR and KL were supported by Medical Research Council programme (MR/P009581/1). This study was also supported by NHLBI (R01 HL172914; U01 HL175383 to XY).

## Author contributions
R.B.-J., D.C. and S.M. performed the experimental work. Y.W., K.C. and Y.X. performed the bioinformatic analysis. M.P. and R.C. helped with laser capture microdissection, R.R. and V.P.K. performed some of the animal experiments, K.L. and E.L.R. contributed AT2 cells and advised on the work. S.R.J. conceived and directed the study, obtained the funding and saw the patients. R.B.-J. and S.R.J. wrote the manuscript with all authors providing intellectual input and approving the final version.

## Competing interests
The authors declare no competing interests.
