## [Transparent Peer Review file · Nature Communications]

mTOR dysregulation induces IL6 and paracrine AT2 cell senescence impeding lung repair in lymphangiomyomatosis

Corresponding Author: Professor Simon Johnson

Version 0:

Reviewer comments:

Reviewer #1

(Remarks to the Author)

This study investigates an important disease of LAM and investigates senescence in LAM microenvironment. The focus on understanding pathogenesis of LAM cysts is interesting and understanding AEC injury repair in this microenvironment is important. While the study has some interesting findings of upregulated markers of senescence in LAM nodules and AECs, coculture and organoid experiments and human cell data, the investigations do not provide a comprehensive insight into a critical role of senescence or IL-6 in LAM disease pathogenesis.

1. Authors demonstrate senescent cells in LAM nodules and AECs. The temporal sequence of events is not clear. Is it a cooccurring phenomenon due to mtorc activation or does one cellular senescence and SASP secretion precedes the other cells senescence?
2. Lack of in vivo model prevents investigation on these mechanisms and also prevents demonstration of a definitive role of IL-6 in the pathogenesis of cysts.

Reviewer #2

(Remarks to the Author)

The present manuscript describes the presence of Senescence markers in LAM lungs which co-localized with alveolar type 2 cells with mTOR inhibition protecting from senescence in patient's tissue. Furthermore, LAM cells grown in organoids altered alveolar type 2 cell growth inducing senescence. The studies demonstrated that IL-6 produced by the LAM cells induced p16 and p21 leading to alveolar type 2 senescence and impaired wound repair. Use of an IL-6 antagonist, Tocilizumab, enhanced the wound repair in TSC2 null LAM cell spheroids. Although the present manuscript describes a potential role for IL-6 in this process and suggests a correlation of IL-6 levels with LAM severity, previous studies have identified cell senescence in LAM tissue associated with mTOR activation and tuberlin loss. Therefore, the presence of senescence in LAM is not novel.

Major Comments:

The number of cells expressing senescence markers in the nodule is incredibly low. Therefore, it is difficult to understand how the senescence is driving impaired healing. Also, it is not clear why the non-LAM cell is accounting for much of the staining for the senescent marker p21. Is this a reflection of p21 expression in macrophages? If so, how do the authors then account for the senescence being induced by mTOR activity from the LAM cell.

Although interesting Figure 3E only represents one human sample in each group. LAM patients exhibit heterogeneity in their phenotype therefore it is unclear if the patient treated with Rapamycin exhibits lower markers of senescence because of the treatment or this is variability in expression between patients. A larger number of samples would need to be studied to make this conclusion.

For the co-cultures shown in Figure 4 were the LAM cells able to induce p21 and p16 expression in the LAFs and vice

versa?

The 621-101/LAF led to increased growth in the AT2 spheroid, therefore, why does this culture demonstrate an increase in the p21 and p16? Wouldn't the opposite effect be expected?

From the data shown it is not clear whether there is a direct relationship between mTOR activation and the cell senescence described.

Reviewer #3

(Remarks to the Author)

• What are the noteworthy results?

- Compared to the lungs of control mice, the pulmonary tumors of the *Tsc2*^{-/-} mouse model have significantly higher senescence-associated beta-galactosidase activity.
- The expression of 125 senescence signature genes of the SenMayo gene panel was significantly enhanced in LAM lung-derived AT2 cells compared to normal lung-derived AT2 cells.
- Characterization of senescent non-proliferative from non-senescent proliferative 621-101 cells in co-culture with LAFs
- The senescence-associated beta-galactosidase activity was induced in 621-101 cells co-cultured with LAM-associated fibroblasts compared to 621-101 monocultures.
- *Tsc2*-add back in 621-103 cells eliminates expression of the p16 (CDKN2A) senescence marker but not the p21 (CDKN1A) marker expression in 621-101 cells.
- There were resolved differences in the growth rates of AT2 cell organoids compared to co-cultures with 621-103/LAFs versus 621-101/LAFs or with the individual cells. Results suggest that induction of senescence in 621-101 and 621-103 cells following co-culture with LAFs was responsible for the reduction in growth rates of AT2 cell organoids.

• Will the work be of significance to the field and related fields? How does it compare to the established literature? If the work is not original, please provide relevant references.

- The work will be of significance to the field in further understanding the role of LAM senescence factors in disease development and progression. However, some methodologies and findings in the manuscript under review are not novel thus dampening enthusiasm for this work. Similar investigations by Bernardelli et al (PMID: 35806041) have linked the secretion of senescence-associated secretory phenotype (SASP) factors to LAM cells and identified lung fibroblasts' role in the process. They also attempted associating mTOR activation via tuberlin loss in LAM to the induction of senescence.

• Does the work support the conclusions and claims, or is additional evidence needed?

- The results provided do not entirely support the claims or conclusions. First, the assumption made in several parts of the manuscript introduction, results, and conclusions that all LAM cell isolates from patient samples exhibit loss of the TSC2 gene has not been proven to be the case in multiple studies (PMID: 26837766, 23504366). Even the findings of the LAM study in which LAM-core cells possibly causative of the LAM phenotype were identified (PMID: 32603599) and that led to the development of the LAM Cell Atlas (PMID: 36599466) cited in this manuscript do not support this assumption; about 40% of LAM-core cells in the publication express TSC2. Laser capture microdissected LAM nodules analyzed in Figure 2D even show that the mTOR pathway was not differently activated to a large degree in this study. Except haploinsufficiency or loss of heterozygosity at the *Tsc2* locus is confirmed in individual LAM cells used in this study, mTOR dysregulation should not be expected to be solely directly causative for the expression of senescence-associated factors in LAM patients.
- An informative addition to the results depicted in Figure 3E could be assessing the expression of p16 in pulmonary tissues of healthy individuals versus LAM patients and those on sirolimus therapy.
- Overall, evidence to support the conclusion that mTOR dysregulation drives secretion of IL6 by LAM cells and, thus, that senescence regulation is TSC2-dependent is rather weak for pulmonary LAM lung tumors but a little stronger for 621-101 cells. Could this difference be tissue type-specific - renal angiomyolipoma vs. pulmonary LAM cells?

• Are there any flaws in the data analysis, interpretation and conclusions? Do these prohibit publication or require revision?

- In interpreting Figure 1A results, the claim that "p21 was most enriched in AT1/AT2 transitional (PATs) cells" (Line 93) was not supported by the figure nor by results in the LAM Cell Atlas. PATs and AT1 cells seem to equally express p21. Another claim in Figure 1A that LAM-core cells were 2-fold enriched in p21 expression compared to other cell types other than macrophages (Lines 94-95) was also not supported by the figure nor by analysis results in the LAM Cell Atlas.
- In Figure 1B, double immunolabeling for cell type-specific markers will enable us to determine which cell types in the representative LAM nodule express p16 or p21. This is especially important as other cell types in the lungs, such as alveolar macrophages, can express these genes.
- The low number of p21 and p16 expressing cells in the representative images shown in Figure 2A, approximately 3% of total LAM nodule cells, makes drawing conclusions regarding senescence marker expression a little difficult. Especially when a complementary image of control lung tissue is not provided. Besides, judging from the image panels in Figure 2A, the percentages of p21 and p16 expressing cells look more like 0.3% of total LAM nodule cells.
- The RNAseq and pathway analysis performed and provided in Figure 2D are reported to help us "understand the

processes driving senescence” (Lines 113 – 119). However, these results report overall differences in LAM versus healthy lung tissue and are not senescence-specific.

- Only very few AT2 cells were observed surrounding the LAM nodule, and many more were in the LAM cyst. However, very few AT2 cells co-express the p21 marker, and thus, conclusions of LAM nodule cell-induced senescence in AT2 cells should be made with healthy skepticism. Besides, healthy lung alveolar macrophages are known to express p21, and one wonders if this was accounted for in your analysis. Besides, some p21-positive cells in LAM cysts in Figure 3A seem to be red blood cells.

- Evidence of cytokine stimulation, apoptosis, and Wnt signaling are typical biochemical pathways in LAM nodules and not directly suggestive of SASP activity. Importantly, were the AT2 cells sorted based on marker expression before the SenMayo Gene Panel analysis? If not, why are the tissues used in the analysis referred to as LAM AT2 cells? Besides, even in the SenMayo gene signature panel, there was some degree of expression of senescence genes in control tissues as depicted in Figure 3C so a statistic quantifying the difference in senescence panels between control and LAM tissue should be provided before definitive conclusions can be made of expression differences using the gene signature panel.

- An image depicting a wider surface area of the cell culture shown in Figure 4A will be more informative regarding the reported cell size and shape differences. Additionally, staining for lysosomes will reveal the increased lysosome number in 621-101 cells upon co-culture with LAFs and thus supports statements in this manuscript. Besides, one of the proliferative 621-101 cells depicted in Figure 4A does show some senescence expression that should at least be discussed.

- The claim of nuclear translocation of p16 in 621-101 cells following co-culture with LAFs was not supported by results depicted in supplementary Figure 4A or 4B.

- Equalizing the Y-axis scaling of the LAM and LAF graphs depicted in Figure 4B would better compare their SA-beta-galactosidase activities.

- The claim that senescence induction is TSC2 dependent using 621-101 and 621-103 cells is not directly supported by the data depicted in Figure 4B because although the bar graphs are lower for 621-103 cells, a statistical comparison was not performed between the two cell types before such conclusions to be made. Additionally, bulk RNAseq results depicted in supplementary Figure 4C and reported to show an induction of senescence-initiating genes in a TSC2-dependent manner (Lines 148-149) was not interpreted accurately. The heatmap depicted in Supplementary Figure 4C of 621-101 vs 621-101/LAF cell co-culture reveals that most of the senescence markers do not show significant differences between groups; only RBL2 was significantly higher in 621-101 cells co-cultured with LAFs.

- The coloring of the bar graphs depicted in Figure 4C and the color palette used in the legend do not match. Also, statistical analysis is usually performed to assess differences between groups (LAF vs 621-101; LAF vs 621-103) compared in Figure 4C, but this was not done. As such, it cannot be confidently stated that beta-galactosidase activity in LAFs truly differs from the co-culture conditions. The same applies to the results depicted in Figure 4D.

- (Lines 196 – 197) A correlation between serum IL6 levels and the 6-minute walking distance can not be made with an R-square of 14%.

• Is the methodology sound? Does the work meet the expected standards in your field?

The methodology employed is sound and meets expected standards in the LAM field.

• Is there enough detail in the methods for the work to be reproduced?

- In some instances, not enough detail was provided in the methods section for the work to be reproduced:

a) The method of deriving AT2 cells from fetal material was not described, and the cell type is key to many conclusions reached in this manuscript, so this is important.

b) Parameters used to predict upstream regulators in ingenuity pathway analysis were not described.

c) The method for preparing lysates from pulmonary tumors of TSC2^{-/-} depicted in Figure 1D was not adequately described. Was only the tumor excised and dissociated for use, or was the whole lung in the mouse model of LAM used? Even though control mice were given sham PBS tail vein injections, was their lung tissue used as the control tissue after 11 weeks? In fact, the animal modeling subsection of the methods section lacks much information.

d) The IL-6 ELISA study was not adequately described. How many replicates per patient were there? The name of the company where the kit was purchased was not adequately provided to necessitate replication.

e) AT2 cell organoid formation and immunofluorescence could be more clearly described.

Version 1:

Reviewer comments:

Reviewer #1

(Remarks to the Author)

The authors have incorporated longitudinal analysis and some murine modeling which strengthens their hypothesis and manuscript.

However, it appears that rapamycin decreased IL6 levels but not senescence. So why do the authors think that targeting IL6 can still be beneficial .

More importantly, not sure why rapamycin + IL6 signaling inhibition was not tested in the new in vivo model. That is important if targeting IL6 is being proposed as a potential therapy.

Reviewer #2

(Remarks to the Author)

The authors have responded to the critiques from the prior review. Issues addressed are summarized below:

The authors describe that the purpose of their work is to establish an association between LAM and senescence and yet such a finding was well reported in Bernardelli et al, which the authors now cite after the first review. The authors argue that their work is more extensive as they examine AT2 cells and look at lung repair with co-cultures and animal models. Furthermore, the authors extensively examine human LAM tissue samples which Bernardelli et al did not.

In response to the prior critiques, using co-immunostaining the authors have shown that although Macrophage/p16 positive cells are present, the macrophages are not within the LAM nodule and most of the senescence marker is not in the CD68 (macrophage) cell.

The use of the scanning microscope system to capture a wider field gives a better view of the senescent cells and improves the manuscript with quantitative data.

Reviewer #3

(Remarks to the Author)

Version 2:

Reviewer comments:

Reviewer #1

(Remarks to the Author)

The authors have satisfactorily addressed my concern and the use of combined Il-6 targeting with Rapamycin in organoids is acceptable.

Reviewer #3

(Remarks to the Author)

Point by point response

Reviewer #1 (Remarks to the Author)

This study investigates an important disease of LAM and investigates senescence in LAM microenvironment. The focus on understanding pathogenesis of LAM cysts is interesting and understanding AEC injury repair in this microenvironment is important. While the study has some interesting findings of upregulated markers of senescence in LAM nodules and AECs, coculture and organoid experiments and human cell data, the investigations do not provide a comprehensive insight into a critical role of senescence or IL-6 in LAM disease pathogenesis.

Point 1.1 *Authors demonstrate senescent cells in LAM nodules and AECs. The temporal sequence of events is not clear. Is it a cooccurring phenomenon due to mTORC activation or does one cellular senescence and SASP secretion precedes the other cells senescence?*

Response 1.1 To answer this question we used two new approaches. Firstly, using an immunocompetent homograft model of LAM, TSC null TTJ cells were injected into the tail vein of albino black six mice and lung tumours were allowed to form over eight weeks. Two weeks following cell or sham injections, animals were treated with the mTORC1 inhibitor rapamycin or vehicle and lung, serum and bronchoalveolar lavage collected at four and eight weeks. Lung tissue was stained for beta-galactosidase, p21 and p16 and co-immunostained with SPC. We observed that beta-galactosidase/SPC, p21/SPC and p16/SPC dual positive AT2 cells were present at both time points but increased in number from four to eight weeks. Both BAL and serum IL6 levels also increased over time. Although IL6 secretion was reduced by treatment with rapamycin, there was only a small and non-significant reduction in senescent cells after rapamycin treatment, suggesting the majority of senescent cells were established within the first two weeks prior to rapamycin therapy and although IL6 secretion could be suppressed, the senescent state was irreversible and these cells persisted. Consistent with this, in a second animal model in which lung cysts formed due to genetic deletion of *Tsc2* in mesenchymal cells: when treated with rapamycin from the start of the study, the number of senescent cells was significantly reduced suggesting senescence was not allowed to develop in the presence of rapamycin. To address this in the human disease, we examined further patient sections stratified into four stages by disease duration and lung function impairment. We compared the distribution of p16 positive LAM cells, LAM associated fibroblasts and AT2 cells by quantifying the percentage of p16 expressing cells as the fraction of that whole cell type. We observed that AT2 cells were most commonly senescent and evident in earlier disease. Using two markers for LAM associated fibroblasts TCF21 and MFAP5, we notice that p16 dual positive fibroblasts were not a feature of very early disease. The number of all senescent cells increased with disease duration but only in advanced disease, categorised by poor lung function and extensive fibroblast infiltration, were PNL2/p16 dual positive cells (senescent LAM cells) seen. The main and initiating event here is the mTOR dependent production of IL6 by the LAM cell causing AT2 cell senescence. Importantly, the evolution of the LAM nodule, particularly the recruitment of LAM associated fibroblasts alters the profile of senescent cells in LAM. The presence of the LAM associated fibroblast allows these cells to become senescent, due to IL6 occurs later in the disease course. This allows the amplification of senescence induction reflected by increasing AT2 cell, fibroblast and eventually, limited LAM cell senescence. These findings have been added to the text, figures, including new cell type specific quantification of senescent cells and an illustrative summary.

Point 1.2 *Lack of in vivo model prevents investigation on these mechanisms and also prevents demonstration of a definitive role of IL-6 in the pathogenesis of cysts.*

Response 1.2 To study the effect of senescence in cyst formation we collaborated with the Krymskaya lab at the University of Pennsylvania. Dr Krymskaya has generated the only animal model of LAM which uses a TBX4 driven mesenchymal-restricted deletion of *Tsc2* in the mouse lung to produce a mTORC1-driven lung cysts associated with decline in pulmonary function rather than TSC null lung tumours. As described above, using this model we observed p16/SPC co-expressing cells in the walls of lung cysts which were suppressed by rapamycin, confirming that loss of mesenchymal *Tsc2* drives AT2 cell senescence which is associated with lung cyst formation. Furthermore, in the TSC2 null cell homograft animals, we were able to measure IL6 in bronchoalveolar lavage fluid and serum where we were able to show time dependent increase in IL6 which was reduced by treatment with rapamycin.

Reviewer 2:

The present manuscript describes the presence of Senescence markers in LAM lungs which co-localized with alveolar type 2 cells with mTOR inhibition protecting from senescence in patient's tissue. Furthermore, LAM cells grown in organoids altered alveolar type 2 cell growth inducing senescence. The studies demonstrated that IL-6 produced by the LAM cells induced p16 and p21 leading to alveolar type 2 senescence and impaired wound repair. Use of an IL-6 antagonist, Tocilizumab, enhanced the wound repair in TSC2 null LAM cell spheroids. Although the present manuscript describes a potential role for IL-6 in this process and suggests a correlation of IL-6 levels with LAM severity, previous studies have identified cell senescence in LAM tissue associated with mTOR activation and tuberlin loss. Therefore, the presence of senescence in LAM is not novel.

Response. We agree that the association of mTOR and senescence is well reported and are aware of the work of Bernardelli et al (PMID: 35806041) which is cited in the revised manuscript. We feel our work significantly adds to the field generally and the findings of Bernadelli and colleagues as the Bernadelli study focuses exclusively on *in vitro* work using TSC2 null cell lines and does not investigate AT2 cells or lung repair. Our work builds upon this by using complex co-cultures, two animal models and our unique deeply phenotyped human disease tissue and linked patient data and serum archive. The work extends the previous findings to relate senescence to lung cyst formation and highlight a novel and potentially tractable role for IL6 in lung damage in LAM.

Major Comments:

Point 2.1 *The number of cells expressing senescence markers in the nodule is incredibly low. Therefore, it is difficult to understand how the senescence is driving impaired healing. Also, it is not clear why the non-LAM cell is accounting for much of the staining for the senescent marker p21. Is this a reflection of p21 expression in macrophages? If so, how do the authors then account for the senescence being induced by mTOR activity from the LAM cell.*

Response 2.1 The number of senescent cells was carefully characterised in deep phenotyped human LAM samples and is similar to the burden of senescent cells in other senescent dependent lung diseases such as idiopathic pulmonary fibrosis (e.g. Schafer et al. Nat Commun 8, 14532 (2017). <https://doi.org/10.1038/ncomms14532>). Of importance to this issue, is that our data, which the revisions strengthen, showing that mTOR driven IL6 secretion is sufficient to initiate the senescence process, rather than the presence of senescent LAM cells which are only observed in more advanced

disease. To make this point clearer, we have performed further co-immunostaining in a larger number of patient samples and used a new slide scanning system to capture a larger area of lung tissue which is more representative of the tissue as a whole. We observe that mTORC1 dysregulation induces LAM cell IL6 secretion which in early disease induces AT2 cell senescence, in later disease where LAM associated fibroblasts are increasingly important components of LAM nodules (Miller S et al. *Journal of Pathology: Clinical Research*. 2020 <https://doi.org/10.1002/cjp2.162>), cross talk between LAF and LAM cells amplifies the senescence response. These data have been added to the manuscript, considered in the discussion and illustrated by a new figure summarising the findings.

To strengthen this work, as suggested by the reviewer, we have extended the co-immunostaining to identify LAM associated fibroblasts and macrophages to determine their contribution to senescence in LAM (see also point 1.1). Analysis of the LAM cell atlas reveals that LAM associated fibroblasts, but not LAM cells or inflammatory cells (the other main components of the LAM nodule) express MFAP5 and have used this to co-localise fibroblast and senescence markers in LAM nodules, giving us a better view of the senescent components of the LAM nodule, and also in response to reviewer one's point, the temporal evolution of these changes. CD68 has been used to identify macrophages and we show that although CD68/p16 positive cells (senescent macrophages) are present, macrophages are not present with LAM nodules and most senescence marker staining is in CD68 negative cells. To highlight this issue, an illustrative figure from 15 individual donors has been added to the supplementary data.

Point 2.2 *Although interesting Figure 3E only represents one human sample in each group. LAM patients exhibit heterogeneity in their phenotype therefore it is unclear if the patient treated with Rapamycin exhibits lower markers of senescence because of the treatment or this is variability in expression between patients. A larger number of samples would need to be studied to make this conclusion.*

Response 2.2 We agree that it would be preferable to study a larger number of samples. For the untreated sample data in the figure this comprises single cell data from 13 subjects, this has been made clear in the figure legend. Unfortunately, LAM tissue from patients treated with rapamycin is extremely unusual. LAM is already a very rare / orphan disease, and biopsy tissue is normally obtained from patients pre-diagnosis and therefore pre-rapamycin treatment. Further in the UK, Europe and much of the USA, although patients may donate explant lung tissue post-transplant, rapamycin is withdrawn on active transplant listing due to concerns over post-operative complications in rapamycin treated patients. This means that mTOR inhibitor treated unpreserved (and also preserved) tissue suitable for single cell sequencing is incredibly unusual, even despite our long-established patient database and LAM sample collection network. We have continued to include this figure as we feel it is a unique proof of concept example, but in response to the reviewer's comment, have added a caveat to the description of this figure.

Point 2.3 *For the co-cultures shown in Figure 4 were the LAM cells able to induce p21 and p16 expression in the LAFs and vice versa?*

Response 2.3 In these experiments we examined senescence associated β -galactosidase expression as a standard marker of senescence. To address the reviewer's question, we have now measured p16 and p21 protein by western blot and immunocytochemistry in both TSC2 null and TSC addback LAM cell and LAF co-cultures. These experiments show that in parallel to the induction of senescence associated β -galactosidase activity p16 and p21 were induced in LAM cells in an mTOR dependent manner. We also examined the effect of IL6 inhibitor, in these experiments and observed that IL6 further

enhanced p16 and p21 expression and was synergistic with mTOR1 signalling. These data have been added to figure 6.

Point 2.4 *The 621-101/LAF led to increased growth in the AT2 spheroid, therefore, why does this culture demonstrate an increase in the p21 and p16? Wouldn't the opposite effect be expected?*

Response 2.4 The data in figure 5 shows that the growth of AT2 cell organoids was initially more rapid than untreated organoids and consistent with the induction of senescence over 14 days AT2 organoid growth was suppressed. The expression of p16 and p21 was examined after the induction of senescence by the LAM nodule cultures after 14 days.

Point 2.5 *From the data shown it is not clear whether there is a direct relationship between mTOR activation and the cell senescence described.*

Response 2.5 Our *in vitro* data (figures 4, 5 and 6) show the induction of senescence associated β -galactosidase, p16 and p21 in fibroblasts and AT2 cells respectively. Suppression of mTOR signalling by genetic addback of TSC2 and the use of rapamycin both reduce canonical markers of senescence. Furthermore, in human lungs, senescence is increased in LAM compared with control tissue (figures 1, 2 and 3). In the new *in vivo* work both *Tsc2* loss and rapamycin affect senescence in mouse lungs (figures 4 and 5). We have therefore used a range of independent, yet complementary approaches which we feel collectively link mTOR dysregulation to senescence in LAM.

Reviewer #3 (Remarks to the Author)

• *What are the noteworthy results?*

- *Compared to the lungs of control mice, the pulmonary tumors of the *Tsc2*^{-/-} mouse model have significantly higher senescence-associated beta-galactosidase activity.*

- *The expression of 125 senescence signature genes of the SenMayo gene panel was significantly enhanced in LAM lung-derived AT2 cells compared to normal lung-derived AT2 cells.*

- *Characterization of senescent non-proliferative from non-senescent proliferative 621-101 cells in co-culture with LAFs*

- *The senescence-associated beta-galactosidase activity was induced in 621-101 cells co-cultured with LAM-associated fibroblasts compared to 621-101 monocultures.*

- **Tsc2*-add back in 621-103 cells eliminates expression of the p16 (CDKN2A) senescence marker but not the p21 (CDKN1A) marker expression in 621-101 cells.*

- *There were resolved differences in the growth rates of AT2 cell organoids compared to co-cultures with 621-103/LAFs versus 621-101/LAFs or with the individual cells. Results suggest that induction of senescence in 621-101 and 621-103 cells following co-culture with LAFs was responsible for the reduction in growth rates of AT2 cell organoids.*

• *Will the work be of significance to the field and related fields? How does it compare to the established literature? If the work is not original, please provide relevant references.*

- The work will be of significance to the field in further understanding the role of LAM senescence factors in disease development and progression. However, some methodologies and findings in the manuscript under review are not novel thus dampening enthusiasm for this work. Similar investigations by Bernardelli et al (PMID: 35806041) have linked the secretion of senescence-associated secretory phenotype (SASP) factors to LAM cells and identified lung fibroblasts' role in the process. They also attempted associating mTOR activation via tuberlin loss in LAM to the induction of senescence.

Response. As stated in the response to reviewer 2, we agree that the association of mTOR and senescence is well reported. Bernardelli and co-workers' paper (PMID: 35806041) focuses exclusively on *in vitro* work to show the co-induction of senescence *in vitro*. Our work significantly adds to this, studying senescence in human disease tissue, in complex co-cultures and in two animal models. The work extends these findings to relate senescence to lung cyst formation and highlight a novel and potentially tractable role for IL6 in lung damage, all of which is novel with respect to lung damage in LAM.

- Does the work support the conclusions and claims, or is additional evidence needed?

Point 3.1 *The results provided do not entirely support the claims or conclusions. First, the assumption made in several parts of the manuscript introduction, results, and conclusions that all LAM cell isolates from patient samples exhibit loss of the TSC2 gene has not been proven to be the case in multiple studies (PMID: 26837766, 23504366). Even the findings of the LAM study in which LAM-core cells possibly causative of the LAM phenotype were identified (PMID: 32603599) and that led to the development of the LAM Cell Atlas (PMID: 36599466) cited in this manuscript do not support this assumption; about 40% of LAM-core cells in the publication express TSC2. Laser capture microdissected LAM nodules analyzed in Figure 2D even show that the mTOR pathway was not differently activated to a large degree in this study. Except haploinsufficiency or loss of heterozygosity at the Tsc2 locus is confirmed in individual LAM cells used in this study, mTOR dysregulation should not be expected to be solely directly causative for the expression of senescence-associated factors in LAM patients.*

Response 3.1 We agree that not all patients with LAM have demonstrable loss of TSC2 although it has not been directly studied in a large number of patients. Furthermore, this depends upon the methods used and tissues studied. For example, analysis of circulating LAM cells demonstrated loss of TSC2 in all 12 patients analysed (Pacheto-Rodrigues et al <https://doi.org/10.1158/0008-5472.CAN-07-1356> and in tissues J46H00u/m20Genet (2002) 47:20–28, <https://doi.org/10.1164/ajrccm.187.6.663>, <https://doi.org/10.1186/s12890-022-02154-0>). Most groups working in the field consider that loss of TSC2 function by some mechanism underlies the pathogenesis of LAM and in our work (figures 4, 5 and 6) genetic addback of TSC2 corrects the induction of senescence. However, the point of the work is to establish an association between LAM and senescence to design new therapies to improve lung repair and the precise mechanism of mTOR dysregulation in all patients with LAM is not the focus and outside scope of the current work.

Point 3.2 *An informative addition to the results depicted in Figure 3E could be assessing the expression of p16 in pulmonary tissues of healthy individuals versus LAM patients and those on sirolimus therapy.*

Response 3.2 We agree with this suggestion but as noted in the response to reviewer 2, whilst although mTOR treated human lung tissue is almost never obtainable, however, by an unusual co-incidence we

obtained a small sample of Everolimus treated LAM explant tissue through our tissue retrieval network and have examined expression of p16 and p21 in this sample. Whilst this was only one sample, we observe that a significant number of senescent cells remain present despite mTOR inhibition, consistent with the irreversible nature of senescence and in keeping with our new murine experimental data. These findings have been added to figure 5.

Point 3.3 Overall, evidence to support the conclusion that mTOR dysregulation drives secretion of IL6 by LAM cells and, thus, that senescence regulation is TSC2-dependent is rather weak for pulmonary LAM lung tumors but a little stronger for 621-101 cells. Could this difference be tissue type-specific - renal angiomyolipoma vs. pulmonary LAM cells?

Response 3.3 Our further work showing the relationship between TSC2 null lung tumour burden, BAL and serum IL6 levels and suppression by rapamycin suggest that mTOR driven IL6 secretion is not a feature of angiomyolipoma only. For completeness we also examined p16 and p21 levels in angiomyolipoma derived from patients with LAM and observed greater levels of p16 and p21 proteins, similar to the levels in LAM lung in angiomyolipoma cells compared with adjacent normal renal parenchyma.

- Are there any flaws in the data analysis, interpretation and conclusions? Do these prohibit publication or require revision?

Point 3.4 In interpreting Figure 1A results, the claim that “p21 was most enriched in AT1/AT2 transitional (PATS) cells” (Line 93) was not supported by the figure nor by results in the LAM Cell Atlas. PATS and AT1 cells seem to equally express p21. Another claim in Figure 1A that LAM-core cells were 2-fold enriched in p21 expression compared to other cell types other than macrophages (Lines 94-95) was also not supported by the figure nor by analysis results in the LAM Cell Atlas.

Response 3.4 The original text describing figure 1A states: ‘p21 was ubiquitously expressed throughout the LAM lung including in TSC null LAM^{core} cells. Within the alveolar epithelial cell population, p21 is most enriched in AT1/AT2 transitional (PATS) cells. p16 expression was lower than p21 across all cell types, although >10% of LAM^{core} cells were p21 positive, representing a 2-fold enrichment compared with other cell types other than macrophages’. We accept that whilst the expression of p21 is similar in PATS and AT1 cells, the p values representing the difference from normal AT1 cells are most significant for PATS, within the overall alveolar epithelial population. LAM^{core} cells also had significant increases in p21 and p16 gene expression, we agree that macrophage subsets, as noted and also mesothelial and ciliated secretory cells had increased senescence markers. We have modified the text to make this clearer.

Point 3.5 In Figure 1B, double immunolabeling for cell type-specific markers will enable us to determine which cell types in the representative LAM nodule express p16 or p21. This is especially important as other cell types in the lungs, such as alveolar macrophages, can express these genes.

Response 3.5 This is an important point, we have now performed additional dual immunostaining with the macrophage marker CD68 and novel markers of LAM associated fibroblasts identified from the LAM cell atlas, TCF21 and MFAP5. Within the LAM nodule, MFAP5 is only expressed by fibroblasts in LAM lung cells and not by LAM^{core} or other nodule cell types. As MFAP5 is secreted for the quantitative co-localisation studies, we used TCF21 / p16 co-staining. Although p16 is present in other mesenchymal

cells including LAM^{core} cells, the number of TCF21 / p16 positive stromal cells within LAM nodules greatly exceeded the PNL2 / p16 dual positive cells consistent with most TCF21 / p16 positive cells being LAM associated fibroblasts. CD68 positive macrophages are seen in LAM sections, some of which also express p16, however within and around LAM nodules, the great majority of p16 expressing cells are CD68 negative. The CD68 data have been discussed in the text and added to the data supplement. We feel these data enhance the clarity of the findings and are grateful for this suggestion.

Point 3.6 *The low number of p21 and p16 expressing cells in the representative images shown in Figure 2A, approximately 3% of total LAM nodule cells, makes drawing conclusions regarding senescence marker expression a little difficult. Especially when a complementary image of control lung tissue is not provided. Besides, judging from the image panels in Figure 2A, the percentages of p21 and p16 expressing cells look more like 0.3% of total LAM nodule cells.*

Response 3.6 The figure shows representative images only for clarity. Extensive and painstaking manual, blinded counting was performed on multiple samples and fields of view was performed to provide an accurate quantification. The reviewer will find the complementary control images in the original within the data supplement (supplementary figure 2). To increase clarity for readers and in response to reviewers 2 and 3, we have now repeated the dual staining and used a scanning microscope system to capture a wider field which we think gives a better view of the distribution of senescent cells. As we have repeated and extended all of the dual immunostaining, we have used this quantification data and added the other cell types to figure 2.

Point 3.7 *The RNAseq and pathway analysis performed and provided in Figure 2D are reported to help us “understand the processes driving senescence” (Lines 113 – 119). However, these results report overall differences in LAM versus healthy lung tissue and are not senescence-specific.*

Response 3.7 For the bulk RNAseq, we analysed both LAM vs Control and LAM-laser-cut vs the rest of the lung. For clarity, we have now represented the data to show genes associated with cell cycle regulation and senescence that are up regulated in LAM-laser-cut vs the rest of the lung. The gene table has been added to the supplementary data.

Point 3.8 *Only very few AT2 cells were observed surrounding the LAM nodule, and many more were in the LAM cyst. However, very few AT2 cells co-express the p21 marker, and thus, conclusions of LAM nodule cell-induced senescence in AT2 cells should be made with healthy scepticism. Besides, healthy lung alveolar macrophages are known to express p21, and one wonders if this was accounted for in your analysis. Besides, some p21-positive cells in LAM cysts in Figure 3A seem to be red blood cells.*

Response 3.8 Figure 3A shows a LAM nodule mostly covered with SPC positive type 2 cells, a site which is not present in normal lungs (as originally described by Matsui et al. DOI: 10.5858/2000-124-1642-HOTIPI) and represents a pathological site for AT2 cells and it is not therefore possible to say whether there are a ‘lot or a few’ at this site. However, I would counter the reviewer’s ‘healthy scepticism’ of the presence of senescent AT2 in LAM by the rigorous quantification of co-expressing SPC/p16 and SPC/p21 positive cells showing that these are elevated in LAM compared with control lungs, the consistent single cell RNA sequencing examining the validated 125 gene SenMayo signature analysis and the replication in novel complex co-cultures where mTOR dysregulated LAM cell fibroblast spheroids (but not TSC2 addback cells) induce p16, p21 and beta galactosidase in human AT2 cell

organoids. Furthermore, we have now added data from an animal model showing the presence of senescent AT2 cells surrounding lung cysts in an mTOR dependent manner. We feel these multiple independent experiments and systems are consistent with the mTOR dependent induction of senescence in AT2 cells in LAM.

Point 3.9 *Evidence of cytokine stimulation, apoptosis, and Wnt signaling are typical biochemical pathways in LAM nodules and not directly suggestive of SASP activity. Importantly, were the AT2 cells sorted based on marker expression before the SenMayo Gene Panel analysis? If not, why are the tissues used in the analysis referred to as LAM AT2 cells? Besides, even in the SenMayo gene signature panel, there was some degree of expression of senescence genes in control tissues as depicted in Figure 3C so a statistic quantifying the difference in senescence panels between control and LAM tissue should be provided before definitive conclusions can be made of expression differences using the gene signature panel.*

Response 3.9 The single cell analysis of AT2 cells was initially stratified by AT2 cell markers as is standard practice in both the lung map and LAM cell atlas analysed by Dr Xu, who is also a member of the lung map team. The term 'LAM AT2 cell' was used for any AT2 cell derived from human LAM lung tissue. The SenMayo gene panel comprises 125 individual genes and is a validated method showing that the overall signature reflects the induction and suppression of senescence in multiple cells and tissue types. The individual genes are normally expressed but the figure shows the overall change in the panel and is a robust and validated way of representing cellular senescence (Saul et al. doi: 10.1101/2021.12.10.472095). In this work we extracted out the AT2 from integrated LAM cell atlas data from LAM lungs (n=10) (PMC10280816) and compared with AT2 cell from female controls, we identified genes differentially expressed (DE) in LAM vs. control in AT2 cells, the statistical criteria to define DE genes are: p-value < 0.05, fold change > 1.5 and expression frequency in AT2 cells \geq 20%. Among these, 365 differentially expressed genes are up-regulated in LAM-AT2 vs. control-AT2 cells, those genes were subjected to the Senmayo geneset analysis (Fig. 3C) and functional enrichment analysis (Fig 3D). In Fig 3C, Pathway signature scores Corresponding uMAPs across LAM and control AT2 were plotted based on Senmayo signature scores which were calculated based on the Senmayo gene expression in LAM and control AT2 using Single-Cell Signature Explorer (PMC6868346). Umap is a visualization tool, the statistical assessment of the Senmayo gene expression in LAM-AT2 vs control-AT2 cell gene expression were present in supplementary table 2 of the supplementary data in the original submission.

Point 3.10 *An image depicting a wider surface area of the cell culture shown in Figure 4A will be more informative regarding the reported cell size and shape differences. Additionally, staining for lysosomes will reveal the increased lysosome number in 621-101 cells upon co-culture with LAFs and thus supports statements in this manuscript. Besides, one of the proliferative 621-101 cells depicted in Figure 4A does show some senescence expression that should at least be discussed.*

Response 3.10 Multiple images of these cells were included in supplementary figure 4 of the original manuscript. We are somewhat constrained by space in the main figure panels, but have now added an expanded panel of the cell culture showing proliferative (nuclear Ki67 positive and minimal SA beta-galactosidase) and senescent (enlarged cells with extensive granular lysosomal SA beta-galactosidase and absent Ki67 staining) LAM cells and have also expanded the image of senescent cells under multiple culture conditions in the supplement. For illustrative purposes we have enlarged the panel (now figure 6A) contrasting proliferative and senescent cells *in vitro* within the space available in the

main manuscript. The new images clearly show the SA beta-galactosidase positive lysosomes increasing in size and number in the TSC2 null compared with TSC2 addback cells. **The co-existence of proliferation and senescence markers has been discussed.**

Point 3.11 *The claim of nuclear translocation of p16 in 621-101 cells following co-culture with LAFs was not supported by results depicted in supplementary Figure 4A or 4B.*

Response 3.11 The nuclear and cytoplasmic lysate fractions show that the TSC2 null 621-101 cells, but not the addback 621-102 or 621-103 cells, both as mono-cultures and co-cultured with LAM associated fibroblasts express nuclear p16. p21 was present in nuclei for all cells under all conditions. These findings are paralleled by the fluorescent immunostaining. I have changed the word 'translocated' to 'expressed' in the manuscript as, the reviewer is correct in that we do not show a dynamic movement into the nucleus but the presence of the protein.

Point 3.12 *Equalizing the Y-axis scaling of the LAM and LAF graphs depicted in Figure 4B would better compare their SA-beta-galactosidase activities.*

Response 3.12 This has been done.

Point 3.13 *The claim that senescence induction is TSC2 dependent using 621-101 and 621-103 cells is not directly supported by the data depicted in Figure 4B because although the bar graphs are lower for 621-103 cells, a statistical comparison was not performed between the two cell types before such conclusions to be made. Additionally, bulk RNAseq results depicted in supplementary Figure 4C and reported to show an induction of senescence-initiating genes in a TSC2-dependent manner (Lines 148-149) was not interpreted accurately. The heatmap depicted in Supplementary Figure 4C of 621-101 vs 621-101/LAF cell co-culture reveals that most of the senescence markers do not show significant differences between groups; only RBL2 was significantly higher in 621-101 cells co-cultured with LAFs.*

Response 3.13 We have now added the statistical analysis of the *in vitro* co-culture experiments (now figure 6). **To support the heatmap we have performed individual quantitative RT PCR analyses of the cell types, the data confirming the heatmap findings has been added to the data supplement.**

Point 3.14 *The coloring of the bar graphs depicted in Figure 4C and the color palette used in the legend do not match. Also, statistical analysis is usually performed to assess differences between groups (LAF vs 621-101; LAF vs 621-103) compared in Figure 4C, but this was not done. As such, it cannot be confidently stated that beta-galactosidase activity in LAFs truly differs from the co-culture conditions. The same applies to the results depicted in Figure 4D.*

Response 3.14 We have added the statistical analysis to these figures and changed the shade of grey in the legend.

Point 3.15 *(Lines 196 – 197) A correlation between serum IL6 levels and the 6-minute walking distance can not be made with an R-square of 14%.*

Response 3.15 Whilst the correlation between serum IL6 and six-minute walk distance in women with LAM was unsurprisingly low considering the multiple factors associated with six-minute walk distance, the association was statistically significant and visual inspection of the data shows those capable of the greatest walk distance had lower values of IL6 than the patients with the lower walk distances. We have now added an additional 30 subjects to the analysis of IL6 and performed a more thorough evaluation of the clinical associations with IL6 in LAM. The extended data and analysis confirm the original findings and also show that elevated IL6 is associated with disease activity, these data have been added to the manuscript and strengthen the relevance of IL6 in LAM.

- *Is the methodology sound? Does the work meet the expected standards in your field?*

The methodology employed is sound and meets expected standards in the LAM field.

- *Is there enough detail in the methods for the work to be reproduced?*

Point 3.16 *In some instances, not enough detail was provided in the methods section for the work to be reproduced:*

a) The method of deriving AT2 cells from fetal material was not described, and the cell type is key to many conclusions reached in this manuscript, so this is important.

b) Parameters used to predict upstream regulators in ingenuity pathway analysis were not described.

c) The method for preparing lysates from pulmonary tumors of TSC2^{-/-} depicted in Figure 1D was not adequately described. Was only the tumor excised and dissociated for use, or was the whole lung in the mouse model of LAM used? Even though control mice were given sham PBS tail vein injections, was their lung tissue used as the control tissue after 11 weeks? In fact, the animal modeling subsection of the methods section lacks much information.

d) The IL-6 ELISA study was not adequately described. How many replicates per patient were there? The name of the company where the kit was purchased was not adequately provided to necessitate replication.

e) AT2 cell organoid formation and immunofluorescence could be more clearly described.

Response 3.16 The amount of experimental detail in the main manuscript is somewhat limited by the word count, and additional methodology are provided in the supplement. The derivation of the AT2 cells was provided in the citation, since the initial submission, a more detailed protocol has been published which is now referenced. The other methodological details have also been expanded in the revised submission.

RE: NCOMMS-24-28827: mTOR dysregulation induces IL6 and paracrine AT2 cell senescence impeding lung repair in lymphangioleiomyomatosis.

Point by point response to reviewers

Reviewer #1 (Remarks to the Author)

The authors have incorporated longitudinal analysis and some murine modeling which strengthens their hypothesis and manuscript.

Thank you we are grateful for these comments.

Point 1.1 *However, it appears that rapamycin decreased IL6 levels but not senescence. So why do the authors think that targeting IL6 can still be beneficial. More importantly, not sure why rapamycin + IL6 signaling inhibition was not tested in the new in vivo model. That is important if targeting IL6 is being proposed as a potential therapy.*

Response 1.1 As senescence is generally regarded as an irreversible process and we agree that neither rapamycin nor IL6 inhibition would reduce the number of existing senescent cells. Our data show that both rapamycin and inhibition of IL6 inhibit the appearance of senescent cells *in vitro* and for rapamycin, also *in vivo*. In new data generated to specifically address this point, we show that the combination of rapamycin and Tocilizumab inhibits the accumulation of the senescence associated protein p21 and to a lesser extent, p16 in a human co-culture system comprising 3D LAM spheroids (LAM derived 621-101 cells and primary LAM associated fibroblast (LAF)) interacting with human AT2 cell organoids over two weeks. We feel this system most closely represents the stromal cell interactions between LAM nodules and alveolar epithelial cells as no *in vivo* model includes the LAM associated fibroblast component of the LAM nodule, meaning animal models for LAM are not currently at the point where *in vivo* models can be reliably predict the response in human disease.

Reviewer #2 (Remarks to the Author)

Point 2.1 *The authors describe that the purpose of their work is to establish an association between LAM and senescence and yet such a finding was well reported in Bernardelli et al, which the authors now cite after the first review. The authors argue that their work is more extensive as they examine AT2 cells and look at lung repair with co-cultures and animal models. Furthermore, the authors extensively examine human LAM tissue samples which Bernardelli et al did not.*

Response 2.1 Our paper reports a long program of work commencing with Medical Research Council funding to study senescence in LAM in 2019, prior to the publication of the Bernadelli report in 2022 and indeed differing from it in key areas as discussed in response to reviewer(s) 3.

Point 2.2 *In response to the prior critiques, using co-immunostaining the authors have shown that although Macrophage/p16 positive cells are present, the macrophages are not within the LAM nodule and most of the senescence marker is not in the CD68 (macrophage) cell. The use of the scanning microscope system to capture a wider field gives a better view of the senescent cells and improves the manuscript with quantitative data.*

Response 2.2 Thankyou, we agree that the macrophage may be an important player in LAM although our new IHC data show it does not appear to be the dominant senescent cell in early disease. We have added a point to the discussion, putting the single cell analysis of transplant lung (figure 1) in context with the protein expression findings in earlier lung disease. We agree that the use of the Akoya scanning microscope can better represent the heterogeneity of the lung tissue in LAM which has added to the manuscript.

Reviewer #3 (Remarks to the Author)

Point 3.1 *Noteworthy Results*

- *The identification of senescent cell accumulation in LAM nodules and lung parenchyma in a TSC2/mTOR-dependent manner.*
- *The demonstration that LAM cell mTOR dysregulation induces IL6 secretion promoting AT2 cell senescence.*
- *The finding that serum IL6 levels correlate with disease severity and lung function decline.*
- *Disease progression involves the accumulation of senescent LAFs that amplify AT2 cell senescence*

Response 3.1 We are grateful for the acknowledgment that our work highlights these key findings and would also add that we show IL6 to be an mTOR dependent SASP protein associated with AT2 cell senescence *in vitro* and have strong correlative mouse and human data from a unique sample set accumulated over many years.

Significance and Originality

Point 3.2 *While the work provides novel insights into LAM pathogenesis, several critical issues limit its impact: The causal relationship between IL6-induced senescence and impaired lung repair remains correlative rather than mechanistic.*

- *The therapeutic implications of targeting IL6 signaling alongside mTOR inhibition lack sufficient experimental validation.*

Response 3.2 We agree with the reviewer that the direct association between IL6 signalling, senescence and lung repair is difficult to prove definitively bearing in mind the widely accepted lack of animal models truly representative of LAM. To circumvent this, we developed a complex human cell culture system as described above using co-culture spheroids in long term culture with human AT2 cell organoids in which to examine senescence and AT2 cell organoid growth as a surrogate for lung repair. We show in the previous work and now new experimental data examining mTOR inhibition, IL6 antagonism and the combination of mTOR and IL6 inhibition, that LAM spheroids increase the presence of p21 in AT2 cells and both rapamycin and Tocilizumab suppress this and that there is synergy between the two drugs, a similar trend is seen for p16.

Point 3.3 *The work deepens some of the findings by Bernardelli et al. (PMID: 35806041), who have linked the secretion of senescence-associated secretory phenotype (SASP) factors to LAM cells and identified the role of lung fibroblasts in the process. They also attempted associating mTOR activation via tuberin loss in LAM with senescence induction and identified IL-8 as an SASP component. However, the similarities in their findings to those in the manuscript under review dampens the originality.*

Response 3.3 Whilst I do not wish to enter into a critique of a separate manuscript, it should be stated for clarity that the Bernadelli paper studies only one cell isolate without evidence of a TSC2 mutation and compares this with a single human lung derived fibroblast culture. Additionally, although reviewer 3 mentions IL8 here, we do not examine IL8 in this paper having previously published on IL8 secretion and induction in LAM derived and normal fibroblasts by LAM derived cells with human validation a year prior to the publication of the Bernadelli paper (Babaei-Jadidi et. al. Mast cell tryptase release contributes to disease progression in lymphangioliomyomatosis. *American Journal of Respiratory and Critical Care Medicine*. 2021 <https://doi.org/10.1164/rccm.202007-2854OC>). Although Bernadelli et. al. show their 'LAM cell' secretes more IL8 than the fibroblast *in vitro*, the differences in isolation methods and characterisation make the significance of this hard to interpret and I feel our work really does advance the idea of senescence, an mTOR dependent SASP and the cell-cell interactions

involved in LAM in a robust way. Further, in the unbiased Cell-Chat analysis in our work highlights the key role of IL6 in LAM cell - AT2 cell interactions.

Point 3.4 - *Although the revised manuscript employs multiple complementary approaches: Human tissue samples and clinical data, two different animal models, in vitro cell culture systems, and advanced techniques including single-cell RNA sequencing, some results obtained were not comprehensively analyzed and do not sufficiently support the hypothesis tested and conclusions, stymying the significance of the results.*

- *The conclusions about senescence driving disease progression rely heavily on correlative data from end-stage disease samples.*

- *The effects of IL6 inhibition on wound repair are only demonstrated in vitro with limited physiological relevance*

- *The data provided did not support some conclusions regarding the expression of senescence markers in LAM and the role of LAM-associated fibroblasts in the process.*

Response 3.4 We clearly show different senescent cell populations increasing over time from the earliest disease at different rates within human LAM and an mTOR dependent relationship with LAM cell and systemic IL6, with IL6 inducing senescence and affecting growth in a state of the art AT2 cell organoid system in keeping with the single cell data. In this orphan disease with very limited access to human tissue no perfect animal model means that complementary and confirmatory data from different methods and sources may be the most robust way to address important clinical questions for patients.

Point 3.5 *Critical data used to draw conclusions are missing, such as images depicting cells co-expressing p16/MFAP5 or co-localization of p16+ and MFAP5+ cells in LAM nodules, or Figure 1D*

Response 3.5 Co-localisation of MFAP5 and p16 are shown in the supplement and we have now added further images to the supplement. The very large numbers of new images requested at revision 1 meant that to fit within journal space constraints, a lot of new corroborating data went into the supplement. There were 43 immunohistochemistry images in the main paper and 72 in the second version of the supplement. We have now incorporated further images into the supplement to address the requests from the two contributors to review 3.

Point 3.6 - *Using only two patient samples for mTOR inhibitor treatment analysis severely limits conclusions about therapy response.*

Response 3.6 As discussed at revision 1, we have added a strong caveat to this point. As mTOR inhibitor treated human tissue is vanishingly uncommon we feel it is worth reporting these findings with the caveats, as they are rare and potentially important adjuncts to the work, allowing the reader to draw their own conclusions.

Point 3.7 - *Lacking the use of proliferation indices to compare senescence-induced phenotypic changes in LAM cell and LAM/LAF co-culture experiments. - Insufficient comparative statistics for senescence marker expression between different cells and cell culture conditions limits the validity of conclusions drawn and information deduced from results.*

Response 3.7 These experiments in figure 6 examine senescence by beta galactosidase expression. Due to the extensive set of cells and conditions used we first examined the co-expression of p16, p21, the proliferation marker Ki67 and beta galactosidase in preliminary experiments to ensure beta galactosidase was a reliable reporter of the senescent phenotype prior to the large screen. For the statistics in these experiments, due to the size of the

experiment and to improve clarity, only the significant, potentially scientific and clinically relevant comparisons are reported.

Point 3.8 - *Specific parameters for cell culture conditions and treatments are inadequately described[1]*

Response 3.8 We have included further details and references in the supplementary methods, in addition to those already provided.

Point 3.9 - *Quantification methods for immunostaining analyses are not standardized as manual cell counting is prone to error and experimenter bias.*

Response 3.9 Where manual counting is used, we have used validated criteria for positive and negative cells with key findings independently checked. Much of the additional data using the Phenolmager Fusion (Akoya Biosciences) and Opal labelling/visualisation quantitated using QuPath software. This has been added to the supplementary methods section.

Point 3.10 - *Statistical analysis approaches for patient stratification are not fully explained*

Response 3.10 The patients studied in figure 3D are stratified on lung function thresholds to reflect disease progression rather than statistics as shown on the lower panel of the figure. Additionally, the FEV₁ thresholds used have now been added to the figure legend.

Reviewer #3 assistant reviewer

Feedback to Key Author Rebuttals to Initial Manuscript Review (Reviewer #3)

Rebuttal discussion 3a.1 *Author Response: As stated in the response to reviewer 2, we agree that the association of mTOR and senescence is well reported. Bernardelli and co-workers' paper (PMID: 35806041) focuses exclusively on in vitro work to show the co-induction of senescence in vitro. Our work significantly adds to this, studying senescence in human disease tissue, in complex co-cultures, and in two animal models. The work extends these findings to relate senescence to lung cyst formation and highlights a novel and potentially tractable role for IL6 in lung damage, all of which is novel with respect to lung damage in LAM.*

Reviewer #3 Feedback: Bernardelli et al (PMID: 35806041) have linked the secretion of senescence-associated secretory phenotype (SASP) factors to LAM cells and identified the role of lung fibroblasts in the process. They also attempted associating mTOR activation via tuberlin loss in LAM with senescence induction. These findings are mimicked to a large extent in the manuscript under review, and some of the alternative models used to deepen knowledge of mTOR-induced senescence in LAM did not yield very convincing results.

Response 3a.1 I don't wish to criticise another published work, but feel I need to correct some of the reviewer's statements. Using exclusively *in vitro* work Bernadelli et. al., by co-culturing two cell types cannot claim that these interactions occur 'in LAM' without studying disease tissue as we did. The *in vitro* changes observed could occur between any two cell types in a tissue culture setting without necessarily being disease relevant. Our results differ from the Bernadelli paper in a number of respects. Firstly, we show using IHC in multiple tissue samples and single cell sequencing that LAM cell senescence is not the initiating event. The single 'LAM cell' used in Bernadelli at. al. was first reported in a paper in 2014 (Lesma et. al. 2014 <https://doi.org/10.1111/jcmm.12237>) and it is possible that the age of this primary culture may have affected its basal senescent state. In contrast we focus upon LAM cell mTOR dependent IL6 rather than IL8 as the main SASP factor (using genetic, *in vivo* and pharmacological means). Using unbiased Cell-Chat analysis of single cell sequencing, we show LAM cell derived IL6, acts on the AT2 cell IL6 receptor. Bernardelli do not mention AT2 cells and investigate only a single candidate SASP factor (IL8 not IL6) in their work. The reviewer's statement that our paper '*did*

not yield very convincing results' is difficult to address specifically, although this does not seem to be the view of the other reviewers.

Rebuttal discussion point 3a.2 Author Response 3.4 The original text describing figure 1A states: 'p21 was ubiquitously expressed throughout the LAM lung including in TSC null LAMcore cells. Within the alveolar epithelial cell population, p21 is most enriched in AT1/AT2 transitional (PATS) cells. p16 expression was lower than p21 across all cell types, although >10% of LAMCORE cells were p21 positive, representing a 2-fold enrichment compared with other cell types other than macrophages'. We accept that whilst the expression of p21 is similar in PATS and AT1 cells, the p values representing the difference from normal AT1 cells are most significant for PATS, within the overall alveolar epithelial population. LAMcore cells also had significant increases in p21 and p16 gene expression, we agree that macrophage subsets, as noted and also mesothelial and ciliated secretory cells had increased senescence markers. We have modified the text to make this clearer.

Reviewer #3 Feedback: The manuscript or supplementary data did not provide evidence of analysis showing a greater significant difference between normal AT1 vs. PATS compared to normal AT1 vs diseased AT1. Analysis showing at least a 2-fold enrichment of p21 expression between AT1 and other cells in the alveolar epithelium was also not provided.

Response 3a.2 Focusing on the current manuscript text which reads 'p21 was ubiquitously expressed including in TSC null LAMcore cells. Within the alveolar epithelial cell population, p21 was most enriched in AT1/AT2 transitional cells (PATS). p16 expression was lower than p21 across all cell types, although >10% of LAMcore cells were p16 positive, a 2-fold enrichment. Senescence markers were also significantly increased in macrophage subsets, mesothelial and ciliated secretory cells (Figure 1A).' We agree that the comparison of LAMcore cells which are not a feature of normal lungs is difficult to conceptualise. Although the plot represents a >2-fold change in p16 in LAMcore cells, we have removed the quantification in case it gives a false impression of the magnitude of p16 gene expression.

Rebuttal discussion point 3a.3 Response 3.6 The figure shows representative images only for clarity. Extensive and painstaking manual, blinded counting was performed on multiple samples and fields of view were performed to provide an accurate quantification. The reviewer will find the complementary control images in the original within the data supplement (supplementary figure 2). To increase clarity for readers and in response to reviewers 2 and 3, we have now repeated the dual staining and used a scanning microscope system to capture a wider field which we think gives a better view of the of the distribution of senescent cells. As we have repeated and extended all of the dual immunostaining, we have used this quantification data and added the other cell types to figure 2.

Reviewer #3 Feedback: Careful dissociation and flow cytometric sorting of tissues will provide a more non-biased, error-free, and accurate comparative quantification of p21, p16, GP100, and PNL2 expressing cells and the co-expressing cells, as opposed to manual cell counting.

Alternatively, software counts could be performed to minimize user error and bias in counting.

Even though the authors stated that rigorous blinding for cell counting experiments was performed, the method of blinding employed was not described.

Additionally, magnified regions of the immunohistochemical images were not indicated in many Figures provided (for example, in Figure 2) in the revised manuscript. As such, one cannot proficiently assess the images presented. Scale bars were also not uniform, and the scaling metric used was not clear in many images provided. This is especially important in images such as Figure 6A, where phenotype size differences due to senescence factors are asserted. As such, even though quantifying the number of the different cell types in LAM nodules provides

further evidence to support the authors' hypothesis, it is important the images provided are in such a manner as to convey the most meaning to a reader of the work.

Response 3a.3 As stated above, we had added further images to the supplement, the missing scale bar has been added to figure 6a. Figure 2 does incorporate magnified panels with 'widefield' being low power view to show tissue architecture and 'nodule' the high-power sections to show single cell level expression. As stated automated image analysis has been incorporated in the new data presented.

We are grateful for the suggestion of using multichannel flow cytometry to unbiasedly evaluate senescent LAM cells. However, we have taken the approach we did for the following reasons. We performed the initial discovery phase using IHC and single cell sequencing. We then used dual label IHC rather than any other method as we could use our unique tissue resource with linked lung function and outcome data so we could relate the development of senescence to disease stage and outcome. The low availability of fresh tissue for flow cytometry in patients with LAM (data from the UK LAM centre shows one lung biopsy and one transplant sample in the UK over last 12 months) that could be analysed by flow cytometry. Furthermore, these samples would not have associated outcome data. Moreover, having worked with LAM lung tissue for many years, we do not think that the cells embedded in the dense extra-cellular matrix within a LAM nodule (Clements et. al. bioRxiv. 2024. <https://doi.org/10.1101/2024.05.16.594484>) could be reliably separated into single cells yet still maintain the protein epitopes necessary for multichannel flow cytometry.

Rebuttal discussion point 3a.4 *Response 3.8 Figure 3A shows a LAM nodule mostly covered with SPC positive type 2 cells, a site which is not present in normal lungs (as originally described by Matsui et al. DOI: 10.5858/2000-124-1642-HOTIPI) and represents a pathological site for AT2 cells and it is not therefore possible to say whether there are a 'lot or a few' at this site. However, I would counter the reviewer's 'healthy scepticism' of the presence of senescent AT2 in LAM by the rigorous quantification of co-expressing SPC/p16 and SPC/p21 positive cells showing that these are elevated in LAM compared with control lungs, the consistent single cell RNA sequencing examining the validated 125 gene SenMayo signature analysis and the replication in novel complex co-cultures where mTOR dysregulated LAM cell fibroblast spheroids (but not TSC2 addback cells) induce p16, p21 and beta galactosidase in human AT2 cell organoids. Furthermore, we have now added data from an animal model showing the presence of senescent AT2 cells surrounding lung cysts in an mTOR dependent manner. We feel these multiple independent experiments and systems are consistent with the mTOR dependent induction of senescence in AT2 cells in LAM.*

Reviewer #3 Feedback: The authors provided statistics in Figure 3A showing the percentage of AT2 cells in the LAM nodule expressing p16 and p21 markers compared to parenchyma and control cells. However, only images of pulmonary LAM p21/SPC co-localization were provided in the manuscript or supplementary data. It is stated in the revised manuscript that the p16/SPC co-localization data is not shown. Given that such images form the basis of the quantification used in the analysis and conclusions drawn, it is necessary to provide sample images of p16/SPC co-localization that correlates with the graph results provided. Also, why weren't side-by-side double-stained p21/SPC "control" lung images provided in the main manuscript.

Response 3a.4 We have now added example p16/SPC dual immunostaining to the data supplement and side-by side control images as requested.

Major Comments

Point 3a.5 *The statement that the number of TCF21 / p16 positive cells within LAM nodules greatly exceeded the PNL2 / p16 dual positive cells consistent with most TCF21 / p16 positive*

cells being LAM-associated fibroblasts is not supported by the sample image data presented in Figure 2A. As such, the quantification results provided in Figure 2B for TCF21/p16 co-localization being significantly higher than the number of p16 expression and about 75% of the number of TCF21 expressing cells is also not supported by the sample immunohistochemical image of TCF21/p16 co-localization.

Besides, statistical comparison of p16 expression between LAM vs. control, or co-localization counts between LAM vs. control were also not performed in Figure 2B. Yet, concrete conclusions are drawn regarding higher p16/TCF21 co-localization levels in LAM compared to control cells. The statistics in the Figures should directly support the conclusions drawn. Additionally, PNL2 stains only the melanocytic cells in LAM and not the smooth muscle cell component, and about 80% of LAM cells are typically PNL2-positive 1,2. Therefore, PNL2/p16 dual staining does not account for all LAM cells in the nodule. As such, one cannot accurately state that the proportion of TCF21/p16 expressing LAM-associated fibroblasts is greater than PNL2/p16 LAM nodule cells

Response 3a.5 As stated in the manuscript, there are no known immunohistochemical markers of fibroblasts. We have previously extensively characterised and shown that the LAM associated fibroblast is the dominant stromal cell within the LAM nodule in more advanced disease (Clements et. al. <https://doi.org/10.1371/journal.pone.0126025> Miller et. al. <https://doi.org/10.1002/cjp2.162>). Therefore, to categorise the non-LAM cell senescent population within the LAM nodule, we interrogated the LAM cell atlas single cell data and evaluated two protein markers, expressed by LAFs and ideally not expressed by LAM or inflammatory cells. We noted both MFAP5 and TCF21 are expressed by LAFs with TCF21 also expressed by LAM cells. As MFAP proved to be extensively matrix bound in more advanced disease making quantification difficult, we quantified TCF21. We accept that PNL2 (and HMB45, we studied both) only stain a subpopulation of LAM cells as is the case for all histological markers and make the point that this combination of markers highlights that a significant proportion of LAM associated fibroblasts express senescence markers (TCF21/p16 positive) and this is significantly greater than either PNL2/p16 or HMB45/p16 dual positive cells (LAM cells), particularly in early disease. Rather than producing somewhat arbitrary and precise figures the data show development of senescence in the LAM nodule, starting with the fibroblast, with LAM cells appearing in more advanced disease. We have clarified this necessary approach in more detail and highlighted the potential limitations within the manuscript.

Point 3a.6 Given that size is a variable used to support the hypothesis being tested in this study, a scale bar should be included in Figure 6A to clarify the size differences between senescent and non-senescent/proliferative cells. There is a clear difference in Ki67 staining between cells in the image, and one Ki67(-) cell appears larger than all other Ki67(+). However, if a summary assertion is to be made regarding senescence-induced changes in cell sizes in Ki67(-) cells, then cell size measurements need to be quantitatively correlated to Ki67-based proliferation, and a single enlarged image would not suffice to make such an assertion.

Additionally, comparative nuclear Ki67 proliferative indices should be calculated between 101 vs. 103 vs. 101/LAF vs. 103/LAF /LAF co-cultures and drug-treated conditions. Only then can the “loss of a proliferative marker” in line 172, or loss of proliferation as a cellular phenotype, be linked to induction of senescence; the postulate cannot be supported by visual evidence alone.

Response 3a.6 The point of the image in figure 6A is to demonstrate the senescent phenotype in LAM cells which are clearly larger, express extensive beta-galactosidase and no nuclear Ki67 compared with the smaller Ki67 positive proliferative cells. We did not feel it necessary to prove senescent cells do not proliferate, are larger than non-senescent cells and express, p21, p16 and beta-galactosidase as this is universally accepted. The point of these figures, as discussed in response to main reviewer 3, was to show using multiple markers that these cells would

become senescent under these conditions and that beta-galactosidase was a suitable readout of this phenotype under these conditions (see point 3.7). For completeness a scale bar has been added to this image.

Point 3a.7 *This statement in lines 141-142 that “the most common senescent cell type was p16/MFAP5 double-positive cells (senescent mesenchymal cells) which increased with worsening disease” cannot be corroborated by the sample results displayed in Figure 3D or the supplementary Figure 4A. Only a few cells in supplementary Figure 4A are MAPF5+ and p16+ doubly-positive for both markers, and as such, the assertion of co-localization of MFAP5+ and p16+ cells in LAM nodules in early disease is not supported.*

Response 3a.7 These illustrative figures are part of a larger quantification as discussed. The nature of reproducing microscopy images is to provide an indication of the quality and distribution of the staining used. We have now added further representative images to the supplementary data, however in a disease with significant heterogeneity it is not possible to show every area evaluated and at some point reviewers and readers do have to believe the results reported as not every aspect of every procedure can be shown.

Point 3a.8 *In lines 149-150, the authors asserted that “ SAβgal/SPC, p21/SPC and p16/SPC dual positive AT2 cells were present at both time points but increased in number from four to eight weeks (Figure 4A & B).” However, there is no immunohistochemical evidence provided in the revised manuscript depicting SAβgal/SPC or p16/SPC double staining in LAM lungs. Additionally, for this assertion to be supported, there is a need to perform a statistical test for the difference in mean dual positive AT2 cell % for each pair of markers for the 4-week vs. 8-week experimental groups to determine if there is an actual increase or decrease in senescence in AT2 cells specifically over time.*

Response 3a.8 Figure 4A contains images of dual labelled SAβgal/SPC, p21/SPC and p16/SPC dual positive AT2 cells. The quantifications and statistics in figure 4B show the presence of SAβgal/SPC, p21/SPC and p16/SPC dual positive AT2 cells at both timepoints as stated in the legend. To make this clearer the y axis label has been changed from (% total cells) to (% total SPC positive cells).

Point 3a.9 *The authors asserted in lines 164-165 that based on immunohistochemistry performed on LAM lung transplant from a patient undergoing everolimus therapy, “...the data is consistent with senescence being mTOR dependent in LAM and irreversible once established.” Besides the need for adequate sample size, notwithstanding limitations associated with sample procurement, there is also a need to include immunohistochemical evidence comparing the expression status of mTOR pathway markers for the drug-treated LAM lung tissue compared with LAM patient biopsies not on sirolimus and healthy lung tissue samples. Only then can such an assertion be made.*

Response 3a.9 The patient from whom the sample was obtained had been treated with mTOR inhibitor over the whole wait time for her transplant listing, with therapeutic levels confirmed by laboratory testing, yet still had significant numbers of senescent cells within the lungs, suggesting that mTOR inhibition at clinical doses will not eliminate established senescent cells.

Minor Comments

Point 3a.10 *Given the authors’ statement in lines 51-52, “ The key pathological feature is the LAM cell, a clone of cells with loss of TSC gene function, most commonly TSC2.”, do all LAM cells or LAMcore cells exhibit loss of TSC2? As there are exceptions to this, it should be clarified when stated in the introduction with appropriate citations.*

Response 3a.10 As stated in the previous round of reviews, in response to the same point made by reviewer 3 (not 3a) most LAM researchers in the field agree with this statement and the evidence tends to support this conclusion, for example, analysis of circulating LAM cells demonstrated loss of TSC2 in all 12 patients analysed (Pacheto-Rodrigues et al <https://doi.org/10.1158/0008-5472.CAN-07-1356>) and in tissues (J46H00u/m20Genet (2002) 47:20–28, <https://doi.org/10.1164/ajrccm.187.6.663>, <https://doi.org/10.1186/s12890-022-02154-0>). Whilst mutations cannot not always be found in LAM samples, this number is increasing with improvements in sequencing technology. To our knowledge, the only other mechanism reported to suppress the TSC2 gene product is epigenetic silencing, reported in one patient with TSC in 2014, coincidentally by some of the authors of the Bernadelli paper (<https://doi.org/10.1111/jcmm.12237>). As stated previously, as this was not the main focus of the paper, we did not discuss this in detail.

Results

Point 3a.11 *In Figure 1B, an image of normal lung parenchyma stained with p21 and p16 is needed next to the LAM-stained images for better visual comparison. Besides, is there any need for Figure 1B in the first place, given that Figure 2A describes the same p16 and p21 staining in LAM nodules? Results in the subsection titled “LAM nodules contain senescent cells” and the subsection titled “Markers of senescence are present in LAM lungs” commonly describe the p16 and p21 expression in LAM nodules and could be combined limiting the tautology of results and providing useful manuscript real estate for figures and writing.*

Response 3a.11 As discussed in the first set of reviews, for space constraints the control pictures are included in the supplement. We felt it was worth also showing the colourimetric visualisation in addition to the fluorescent staining as these images give a better overview of tissue architecture.

Point 3a.12 *Only 2-3 pairs of CD68+ and p16+ cells appear to co-localize or express both markers in Supplementary Figure 4B, and thus, the assertion in lines 112-114 of “a small population of CD68/p16 expressing cells” within the LAM lung parenchyma was not entirely supported.*

Response 3a.12 This figure illustrates that senescent macrophages are not present in LAM nodules and also not a major part of early disease in response to reviewer 2 who accepts this point (see comment 2.2).

Point 3a.13 *In Line 102, a reference to Figure 1C & D is made. Figure 1D does not exist in the revised manuscript.*

Response 3a.13 Thank you for pointing this out, figure 1D was removed in the first round of reviews, this has now been corrected.

Point 3a.14 *There is a need to clarify that the ‘LAM and normal lung-derived AT2 cells’ used in the SenMayo gene signature analysis were resolved from in silico data culled from the LAM Cell Atlas. This will distinguish results from actual tissue-derived AT2 cells used in organoids described in Figures 6 & 7.*

Response 3a.14 This has been clarified in the text.

Point 3a.15 *If scRNAseq of a rapamycin-treated lung was performed, it was not described in the methods or supplementary section. Only analysis of LAM Cell Atlas scRNAseq data was described. There is a need for clarification here.*

Response 3a.15 The rapamycin treated sample was analysed in Dr Xu's laboratory using the same methods as the LAM atlas. This data was shared as part of a collaboration and is not available on the public facing LAM cell atlas at present although the non-rapamycin treated data from this figure is. This has now been clarified in the methods.

Point 3a.16 *In Line 106, Supplementary Figure 2 is referenced rather than supplementary Figure 4A.*

Response 3a.16 Thank you for pointing this out, figure 1D was removed in the first round of reviews, this has now been corrected to Figure 2A and supplementary figure 4.

Point 3a.17 *In Supplementary Figure 5A the top right panel is incorrectly labeled as p16 instead of p21.*

Response 3a.13 Thankyou for pointing this out, it has now been corrected.

Point 3a.18 *RNAseq analysis of laser capture micro-dissected LAM nodules yielded results that add little support to the author's hypothesis; can any senescence-associated genes and pathways be resolved from the data and did it inform any other analysis performed for the study? If not, and it is not readily obvious, consider relegating Figures 2C & D to the supplementary section along with its data, which could rather be presented as the top 10 significantly expressed genes in each pathway identified in the current figure 2D. This will allow more manuscript space for relevant data.*

Response 3a18. We agree that the data examining the transcriptome of 19 laser captured samples does not show a strong senescent signal. This may be due to the relatively small number of senescent cells captured, with the main signals being cell injury and the response to this. Despite this, we think this is worthy of inclusion due to the value of the data overall to the field.

Point 3a.19 *There is a need to describe the legends associated with Figure 3B for better comprehension of analysis results. For instance, how does the legend term "level" apply to the result shown in Figure 3B? Is the colored legend a measure of the number of LAM Cell Atlas samples used in the analysis? There is a need for some clarification of the chart's legend.*

Response 3a19. The coloured bar represents the module score calculated based on the SenMayo genes expression in the AT2 cells. Specifically, the score of the SenMayo gene set in the AT2 cell was computed as the sum of all UMI for all the SenMayo genes expressed in AT2 cells, divided by the sum of all UMI expressed in AT2 cells. The grey bar represents Density Distribution Score which quantifies the density of cells expressing the SenMayo geneset within specific regions of the plot. The score is calculated by dividing the plot into a grid and counting the number of cells within each grid cell. Higher scores indicate regions with a higher concentration of cells expressing the gene signature¹. The blue bar represents the normalized expression levels. We have both simplified the annotation just to express the module score for clarity and enlarged the description of panel 3B in the legend.

Point 3a.20 *In Figure 5B, were the percentages of dual-labeled AT2 cells calculated from the total number of lung cells or all SPC+ cells? The Figure 5 legend states that it is calculated from the total number of lung cells. If so, what is the reason for the "(%SPC positive)" label on the Y-axis of the graphs?*

Response 3a.20 This was addressed in point in point 3a4.

Point 3a.21 *The large differences in the number of LAM patients vs. control or rapamycin-treated serum samples used in comparing IL-6 expression, as depicted in Figure 8F, could bias statistical analysis results to a great extent.*

Response 3a.21 We were careful to match the controls for age and sex and although we were able to test more LAM patients than control subjects we used a statistical test tolerant of unequal sample sizes and as most women with LAM had higher serum IL6 levels than the healthy control women, the result was very strongly statistically significant.

Point 3a.22 *Authors need to specify the sex and gender considerations of animals used in analyses in the methods or results section.*

Response 3a.22 This has been done

mTOR dysregulation induces IL6 and paracrine AT2 cell senescence impeding lung repair in lymphangioliomyomatosis.

Point by point response to reviewers

Reviewer #1 (Remarks to the Author)

The authors have satisfactorily addressed my concern and the use of combined IL-6 targeting with Rapamycin in organoids is acceptable.

Response Thank you

Reviewer #2 (Remarks to the Author)

The authors have responded to the critiques and clarified many details, the statistical methods, and figure legends to improve the manuscript. Furthermore, data regarding the therapeutic implications of targeting IL6 and the senescent cell populations are demonstrated along with co-localization of MFAP5 and p16 (with added images in the supplement). They have appropriately removed the p16 gene expression data and clarified the TCF21/p16, PNL2/p16 data.

With regards to Point 3a18, it is not clear why the authors insist on putting this data into the main manuscript when they themselves agree that it does not contribute to the conclusions of the manuscript. It can be placed in the supplement.

Response Thank you for these helpful remarks. We have elected to leave the laser capture scRNAseq data in figure 2. Whilst we agree that the methodology used didn't add significantly to the overall conclusions, as the transcriptome linked to lung function and outcome data is a valuable resource for the community and may not get the full exposure if exclusively in the supplement. The data included in the main text represents only two panels of a single multi-part figure.

Prompts

- What are the noteworthy results?
- Will the work be of significance to the field and related fields? How does it compare to the established literature? If the work is not original, please provide relevant references.
- Does the work support the conclusions and claims, or is additional evidence needed?
- Are there any flaws in the data analysis, interpretation and conclusions? Do these prohibit publication or require revision?
- Is the methodology sound? Does the work meet the expected standards in your field?
- Is there enough detail provided in the methods for the work to be reproduced?

After carefully reviewing the revised manuscript "mTOR dysregulation induces IL6 and paracrine AT2 cell senescence impeding lung repair in lymphangiomyomatosis," my summarized comments are below. Detailed comments are also attached.

Noteworthy Results

- The identification of senescent cell accumulation in LAM nodules and lung parenchyma in a TSC2/mTOR-dependent manner.
- The demonstration that LAM cell mTOR dysregulation induces IL6 secretion promoting AT2 cell senescence.
- The finding that serum IL6 levels correlate with disease severity and lung function decline.
- Disease progression involves the accumulation of senescent LAFs that amplify AT2 cell senescence

Significance and Originality

While the work provides novel insights into LAM pathogenesis, several critical issues limit its impact:

- The causal relationship between IL6-induced senescence and impaired lung repair remains correlative rather than mechanistic
- The therapeutic implications of targeting IL6 signaling alongside mTOR inhibition lack sufficient experimental validation.
- The work deepens some of the findings by Bernardelli et al. (PMID: 35806041), who have linked the secretion of senescence-associated secretory phenotype (SASP) factors to LAM cells and identified the role of lung fibroblasts in the process. They also attempted associating mTOR activation via tuberlin loss in LAM with senescence induction and identified IL-8 as an SASP component. However, the similarities in their findings to those in the manuscript under review dampens the originality.
- Although the revised manuscript employs multiple complementary approaches: Human tissue samples and clinical data, two different animal models, in vitro cell culture systems, and advanced techniques including single-cell RNA sequencing, some results obtained were not comprehensively analyzed and do not sufficiently support the hypothesis tested and conclusions, stymying the significance of the results.

Data Support and Additional Evidence Needed

- The conclusions about senescence driving disease progression rely heavily on correlative data from end-stage disease samples.
- The effects of IL6 inhibition on wound repair are only demonstrated in vitro with limited physiological relevance
- The data provided did not support some conclusions regarding the expression of senescence markers in LAM and the role of LAM-associated fibroblasts in the process.
- Critical data used to draw conclusions are missing, such as images depicting cells co-expressing p16/MFAP5 or co-localization of p16+ and MFAP5+ cells in LAM nodules, or Figure 1D

Methodological Limitations

- Using only two patient samples for mTOR inhibitor treatment analysis severely limits conclusions about therapy response.
- Lacking the use of proliferation indices to compare senescence-induced phenotypic changes in LAM cell and LAM/LAF co-culture experiments
- Insufficient comparative statistics for senescence marker expression between different cells and cell culture conditions limits the validity of conclusions drawn and information deduced from results.

Reproducibility Concerns

- Specific parameters for cell culture conditions and treatments are inadequately described[1]
- Quantification methods for immunostaining analyses are not standardized as manual cell counting is prone to error and experimenter bias.
- Statistical analysis approaches for patient stratification are not fully explained

Feedback to Key Author Rebuttals to Initial Manuscript Review (Reviewer #3)

- 1) Author Response:** As stated in the response to reviewer 2, we agree that the association of mTOR and senescence is well reported. Bernardelli and co-workers' paper (PMID: 35806041) focuses exclusively on *in vitro* work to show the co-induction of senescence *in vitro*. Our work significantly adds to this, studying senescence in human disease tissue, in complex co-cultures, and in two animal models. The work extends these findings to relate senescence to lung cyst formation and highlights a novel and potentially tractable role for IL6 in lung damage, all of which is novel with respect to lung damage in LAM.

Reviewer #3 Feedback: Bernardelli et al (PMID: 35806041) have linked the secretion of senescence-associated secretory phenotype (SASP) factors to LAM cells and identified the role of lung fibroblasts in the process. They also attempted associating mTOR activation via tuberin loss in LAM with senescence induction. These findings are mimicked to a large extent in the manuscript under review, and some of the alternative models used to deepen knowledge of mTOR-induced senescence in LAM did not yield very convincing results.

- 2) Author Response 3.4** The original text describing figure 1A states: 'p21 was ubiquitously expressed throughout the LAM lung including in TSC null LAMcore cells. Within the alveolar epithelial cell population, p21 is most enriched in AT1/AT2 transitional (PATS) cells. p16 expression was lower than p21 across all cell types, although >10% of LAMCORE cells were p21 positive, representing a 2-fold enrichment compared with other cell types other than macrophages'. We accept that whilst the expression of p21 is similar in PATS and AT1 cells, **the p values representing the difference from normal AT1 cells are most significant for PATS, within the overall alveolar epithelial population.** LAMcore cells also had significant increases in p21 and p16 gene expression, we agree that macrophage subsets, as noted and also mesothelial and ciliated secretory cells had increased senescence markers. We have modified the text to make this clearer.

Reviewer #3 Feedback: The manuscript or supplementary data did not provide evidence of analysis showing a greater significant difference between normal AT1 vs. PATS compared to normal AT1 vs diseased AT1. Analysis showing at least a 2-fold enrichment of p21 expression between AT1 and other cells in the alveolar epithelium was also not provided.

- 3) Response 3.6** The figure shows representative images only for clarity. Extensive and painstaking manual, blinded counting was performed on multiple samples and fields of view were performed to provide an accurate quantification. The reviewer will find the complementary control images in the original within the data supplement (supplementary figure 2). To increase clarity for readers and in response to reviewers 2 and 3, we have now repeated the dual staining and used a scanning microscope system to capture a wider field which we think gives a better view of the of the distribution of senescent cells. As we have repeated and extended all of the dual immunostaining, we have used this quantification data and added the other cell types to figure 2.

Reviewer #3 Feedback: Careful dissociation and flow cytometric sorting of tissues will provide a more non-biased, error-free, and accurate comparative quantification of p21, p16, GP100, and PNL2 expressing cells and the co-expressing cells, as opposed to manual cell counting. Alternatively, software counts could be performed to minimize user error and bias in counting. Even though the authors stated that rigorous blinding for cell counting experiments was performed, the method of blinding employed was not described.

Additionally, magnified regions of the immunohistochemical images were not indicated in many Figures provided (for example, in Figure 2) in the revised manuscript. As such, one cannot proficiently assess the images presented. Scale bars were also not uniform, and the scaling metric used was not clear in many images provided. This is especially important in images such as Figure 6A, where phenotype size differences due to senescence factors are asserted. As such, even though quantifying the number of the different cell types in LAM nodules provides further evidence to support the authors' hypothesis, it is important the images provided are in such a manner as to convey the most meaning to a reader of the work.

- 4) Response 3.8** Figure 3A shows a LAM nodule mostly covered with SPC positive type 2 cells, a site which is not present in normal lungs (as originally described by Matsui et al. DOI: 10.5858/2000-124-1642-HOTIPI) and represents a pathological site for AT2 cells and it is not therefore possible to say whether there are a 'lot or a few' at this site. However, I would counter the reviewer's 'healthy scepticism' of the presence of senescent AT2 in LAM by the rigorous quantification of co-expressing SPC/p16 and SPC/p21 positive cells showing that these are elevated in LAM compared with control lungs, the consistent single cell RNA sequencing examining the validated 125 gene SenMayo signature analysis and the replication in novel complex co-cultures where mTOR dysregulated LAM cell fibroblast spheroids (but not TSC2 addback cells) induce p16, p21 and beta galactosidase in human AT2 cell organoids. Furthermore, we have now added data from an animal model showing the presence of senescent AT2 cells

surrounding lung cysts in an mTOR dependent manner. We feel these multiple independent experiments and systems are consistent with the mTOR dependent induction of senescence in AT2 cells in LAM.

Reviewer #3 Feedback: The authors provided statistics in **Figure 3A** showing the percentage of AT2 cells in the LAM nodule expressing p16 and p21 markers compared to parenchyma and control cells. However, only images of pulmonary LAM p21/SPC co-localization were provided in the manuscript or supplementary data. It is stated in the revised manuscript that the p16/SPC co-localization data is not shown. Given that such images form the basis of the quantification used in the analysis and conclusions drawn, it is necessary to provide sample images of p16/SPC co-localization that correlates with the graph results provided. Also, why weren't side-by-side double-stained p21/SPC "control" lung images provided in the main manuscript for comparison rather than it being placed in **Supplementary Figure 3**. It will add more clarity to **Figure 3A**.

Major Comments

- 1) The statement that the number of TCF21 / p16 positive cells within LAM nodules greatly exceeded the PNL2 / p16 dual positive cells consistent with most TCF21 / p16 positive cells being LAM-associated fibroblasts is not supported by the sample image data presented in **Figure 2A**. As such, the quantification results provided in **Figure 2B** for TCF21/p16 co-localization being significantly higher than the number of p16 expression and about 75% of the number of TCF21 expressing cells is also not supported by the sample immunohistochemical image of TCF21/p16 co-localization.

Besides, statistical comparison of p16 expression between LAM vs. control, or co-localization counts between LAM vs. control were also not performed in **Figure 2B**. Yet, concrete conclusions are drawn regarding higher p16/TCF21 co-localization levels in LAM compared to control cells. The statistics in the Figures should directly support the conclusions drawn.

Additionally, PNL2 stains only the melanocytic cells in LAM and not the smooth muscle cell component, and about 80% of LAM cells are typically PNL2-positive^{1,2}. Therefore, PNL2/p16 dual staining does not account for all LAM cells in the nodule. As such, one cannot accurately state that the proportion of TCF21/p16 expressing LAM-associated fibroblasts is greater than PNL2/p16 LAM nodule cells

- 2) Given that size is a variable used to support the hypothesis being tested in this study, a scale bar should be included in **Figure 6A** to clarify the size differences between senescent and non-senescent/proliferative cells. There is a clear difference in Ki67 staining between cells in the image, and one Ki67(-) cell appears larger than all other Ki67(+). However, if a summary assertion is to be made regarding senescence-induced changes in cell sizes in Ki67(-) cells, then cell size measurements need to be quantitatively correlated to Ki67-based proliferation, and a single enlarged image would not suffice to make such an assertion.

Additionally, comparative nuclear Ki67 proliferative indices should be calculated between 101 vs. 103 vs. 101/LAF vs. 103/LAF /LAF co-cultures and drug-treated conditions. Only then can the "loss of a proliferative marker" in line 172, or loss of proliferation as a cellular phenotype, be linked to induction of senescence; the postulate cannot be supported by visual evidence alone.

- 3) This statement in lines 141-142 that "the most common senescent cell type was p16/MFAP5 double-positive cells (senescent mesenchymal cells) which increased with worsening disease" cannot be corroborated by the sample results displayed in **Figure 3D** or the **supplementary Figure 4A**. Only a few cells in **supplementary Figure 4A** are MAPF5+ and p16+ doubly-positive for both markers, and as such, the assertion of co-localization of MFAP5+ and p16+ cells in LAM nodules in early disease is not supported.
- 4) In lines 149-150, the authors asserted that "SA β gal/SPC, p21/SPC and p16/SPC dual positive AT2 cells were present at both time points but increased in number from four to eight weeks (**Figure 4A & B**)." However, there is no immunohistochemical evidence provided in the revised manuscript depicting SA β gal/SPC or p16/SPC double staining in LAM lungs. Additionally, for this assertion to be supported, there is a need to perform a statistical test for the difference in mean dual positive AT2 cell % for each pair of markers for the 4-week vs. 8-week experimental groups to determine if there is an actual increase or decrease in senescence in AT2 cells specifically over time.

- 5) The authors asserted in lines 164-165 that based on immunohistochemistry performed on LAM lung transplant from a patient undergoing everolimus therapy, "...the data is consistent with senescence being mTOR dependent in LAM and irreversible once established." Besides the need for adequate sample size, notwithstanding limitations associated with sample procurement, there is also a need to include immunohistochemical evidence comparing the expression status

of mTOR pathway markers for the drug-treated LAM lung tissue compared with LAM patient biopsies not on sirolimus and healthy lung tissue samples. Only then can such an assertion be made.

Minor Comments

Introduction

- 6) Given the authors' statement in lines 51-52, "*The key pathological feature is the LAM cell, a clone of cells with loss of TSC gene function, most commonly TSC2.*", do all LAM cells or LAM^{CORE} cells exhibit loss of TSC2? As there are exceptions to this, it should be clarified when stated in the introduction with appropriate citations.

Results

- 7) In **Figure 1B**, an image of normal lung parenchyma stained with p21 and p16 is needed next to the LAM-stained images for better visual comparison. Besides, is there any need for **Figure 1B** in the first place, given that **Figure 2A** describes the same p16 and p21 staining in LAM nodules? Results in the subsection titled "LAM nodules contain senescent cells" and the subsection titled "Markers of senescence are present in LAM lungs" commonly describe the p16 and p21 expression in LAM nodules and could be combined limiting the tautology of results and providing useful manuscript real estate for figures and writing.
- 8) Only 2-3 pairs of CD68+ and p16+ cells appear to co-localize or express both markers in **Supplementary Figure 4B**, and thus, the assertion in lines 112-114 of "*a small population of CD68/p16 expressing cells*" within the LAM lung parenchyma was not entirely supported.
- 9) In Line 102, a reference to Figure 1C & D is made. Figure 1D does not exist in the revised manuscript.
- 10) There is a need to clarify that the 'LAM and normal lung-derived AT2 cells' used in the SenMayo gene signature analysis were resolved from in silico data culled from the LAM Cell Atlas. This will distinguish results from actual tissue-derived AT2 cells used in organoids described in **Figures 6 & 7**.
- 11) If scRNAseq of a rapamycin-treated lung was performed, it was not described in the methods or supplementary section. Only analysis of LAM Cell Atlas scRNAseq data was described. There is a need for clarification here.
- 12) In Line 106, **Supplementary Figure 2** is referenced rather than supplementary **Figure 4A**.
- 13) In **Supplementary Figure 5A** the top right panel is incorrectly labeled as p16 instead of p21.
- 14) RNAseq analysis of laser capture micro-dissected LAM nodules yielded results that add little support to the author's hypothesis; can any senescence-associated genes and pathways be resolved from the data and did it inform any other analysis performed for the study? If not, and it is not readily obvious, consider relegating **Figures 2C & D** to the supplementary section along with its data, which could rather be presented as the top 10 significantly expressed genes in each pathway identified in the current **figure 2D**. This will allow more manuscript space for relevant data.
- 15) There is a need to describe the legends associated with **Figure 3B** for better comprehension of analysis results. For instance, how does the legend term "level" apply to the result shown in **Figure 3B**? Is the colored legend a measure of the number of LAM Cell Atlas samples used in the analysis? There is a need for some clarification of the chart's legend.
- 16) In **Figure 5B**, were the percentages of dual-labeled AT2 cells calculated from the total number of lung cells or all SPC+ cells? The **Figure 5** legend states that it is calculated from the total number of lung cells. If so, what is the reason for the "(%SPC positive)" label on the Y-axis of the graphs?
- 17) The large differences in the number of LAM patients vs. control or rapamycin-treated serum samples used in comparing IL-6 expression, as depicted in **Figure 8F**, could bias statistical analysis results to a great extent.
- 18) Authors need to specify the sex and gender considerations of animals used in analyses in the methods or results section.

References

- 1 Zhe, X. & Schuger, L. Combined smooth muscle and melanocytic differentiation in lymphangioliomyomatosis. *J Histochem Cytochem* **52**, 1537-1542 (2004).
<https://doi.org:10.1369/jhc.4A6438.2004>

2 Krymskaya, V. P. Smooth muscle-like cells in pulmonary lymphangioleiomyomatosis. Proceedings of the American Thoracic Society **5**, 119-126 (2008). <https://doi.org/10.1513/pats.200705-061VS>